# Lysosomal exocytosis releases pathogenic α-synuclein species from neurons in synucleinopathy models

Ying Xue Xie [1,4], Nima N. Naseri[2,4], Jasmine Fels[1], Parinati Kharel[1], Yoonmi Na[1], Diane Lane [3], Jacqueline Burré [1] & Manu Sharma [1] ✉

Considerable evidence supports the release of pathogenic aggregates of the neuronal protein α-Synuclein (αSyn) into the extracellular space. While this release is proposed to instigate the neuron-to-neuron transmission and spread of αSyn pathology in synucleinopathies including Parkinson's disease, the molecular-cellular mechanism(s) remain unclear. To study this, we generated a new mouse model to specifically immunoisolate neuronal lysosomes, and established a long-term culture model where αSyn aggregates are produced within neurons without the addition of exogenous fibrils. We show that neuronally generated pathogenic species of αSyn accumulate within neuronal lysosomes in mouse brains and primary neurons. We then find that neurons release these pathogenic αSyn species via SNARE-dependent lysosomal exocytosis. The released aggregates are non-membrane enveloped and seeding-competent. Additionally, we find that this release is dependent on neuronal activity and cytosolic $Ca^{2+}$. These results propose lysosomal exocytosis as a central mechanism for the release of aggregated and degradation-resistant proteins from neurons.

Cytoplasmic aggregates of the synaptic protein α-synuclein (αSyn) are a characteristic feature of multiple neurodegenerative diseases termed "synucleinopathies", including Parkinson disease (PD). Over the course of these age-linked diseases, αSyn assembles into amyloid-type β-sheet-rich aggregates, depositing as Lewy bodies and/or Lewy neurites within neurons[1–3]. The neuro-anatomical propagation of αSyn pathology is highly stereotyped, and has been defined well-enough to be used in staging PD—originating in the medulla and olfactory bulb, then advancing to the midbrain and basal forebrain, followed by "spread" to the neocortex[4,5]. This observation, combined with detection of extracellular αSyn in human cerebrospinal fluid[6], led to the original conjecture that pathogenic species of αSyn may be transmitted neuron-to-neuron via prion-like permissive templating. Subsequently, in PD patients who had received fetal neuron transplants, appearance of Lewy pathology in the non-diseased grafted neurons also pointed to

the transmission of αSyn pathology from the surrounding PD-affected neurons[7,8]. These observations were followed by animal studies which confirmed either neuron-to-neuron transmission of αSyn pathology[9], or induction of intraneuronal αSyn pathology by an inoculum of extracellular αSyn fibrils[10,11].

However, unclarity persists in how the cytosolic aggregates are conveyed into the extracellular milieu: Proposed pathway(s) include secretion of αSyn monomers, oligomers and/or larger aggregates inside extracellular vesicles—such as exosomes[12–14], or microvesicles[15]—which have been categorized collectively as "unconventional secretion". Other proposed mechanisms include synaptic vesicle release[16], or exocytosis of other types of vesicles[17–19], exophagy[20], membrane translocation[21], or trafficking of lysosome-associated aggregates through tunneling nanotubes[22,23]. Yet, these studies have not identified a neuronal organelle that first accumulates and then secretes the

[1]Appel Institute for Alzheimer's Research, and Feil Family Brain & Mind Research Institute, Weill Cornell Medicine, New York, NY, USA. [2]Department of Chemistry, University of Pennsylvania, Philadelphia, PA, USA. [3]Feil Family Brain & Mind Research Institute, Weill Cornell Medicine, New York, NY, USA. [4]These authors contributed equally: Ying Xue Xie, Nima N. Naseri. ✉e-mail: mas2189@med.cornell.edu

pathogenic αSyn aggregates generated within the neurons, resulting in the current mechanistic gap.

Interestingly, PD is associated with mutations in the glucocer-ebrosidase gene, which cause Gaucher disease, the most common lysosomal storage disorder[24,25]. In addition, multiple cellular pathways have linked PD and αSyn aggregation with lysosomal storage disorders[26,27].

Here, we show that neuronally-produced pathogenic αSyn species accumulate within the neuronal lysosomes in mouse brains and in primary neurons, and that these seeding-competent species are then released into the extracellular space via SNARE-dependent lysosomal exocytosis.

## Results

### Pathogenic species of αSyn accumulate within the lysosomes of transgenic αSyn^A53T mouse brains

Rapid neurodegeneration and synapse loss in cysteine string protein-α knockout mice (CSPα[−/−]) is rescued by transgenic expression of human αSyn carrying the familial PD mutation A53T (Tg-αSyn^A53T)[28]. However, beyond 5 months, the rescued Tg-αSyn^A53T/CSPα[−/−] mice begin exhibiting neurodegeneration due to the transgenic overexpression of αSyn^A53T. Curiously, we found that Tg-αSyn^A53T/CSPα[−/−] mice have accelerated accumulation of the pathogenic versions/species of αSyn compared to Tg-αSyn^A53T/CSPα[+/−] mice, detectable by antibodies against phospho-Ser129 αSyn, filamentous αSyn, and amyloid-type folds (Supplementary Fig. S1). Meanwhile, there was no change in monomeric αSyn levels (Supplementary Fig. S1). CSPα[+/−] mice have normal CSPα function[29]. This suggested that brains of Tg-αSyn^A53T/CSPα[−/−] mice are collecting αSyn aggregates faster in the absence of CSPα.

Loss-of-function mutations in CSPα cause the lysosomal storage disorder adult-onset neuronal ceroid lipofuscinosis (ANCL) or Kufs disease[30–33]. Specifically, loss of CSPα function leads to progressive accumulation of lysosomes containing undegraded material such as lipofuscin/residual bodies in ANCL patient brains[30], in CSPα[−/−] mouse brains[29], in Drosophila models carrying the ANCL mutations[34], and in ANCL patient-derived induced neurons[35]. Tg-αSyn^A53T/CSPα[−/−] mice also accrue lysosomal proteins such as LAMP1 and cathepsin-L, as well as ATP5G—a mitochondrial protein characteristically stored in lyso-somes of CSPα loss-of-function models and ANCL patient neurons (Supplementary Fig. S1). The enhanced accumulation of lysosomal contents in Tg-αSyn^A53T/CSPα[−/−] brains, together with the exaggerated buildup of αSyn aggregates in these brains, plus the association of lysosomal dysfunction and lysosomal storage disorders with PD[24–27], led us to investigate whether neuronally-produced αSyn aggregates deposit within the lysosomes.

We next aimed to determine whether αSyn aggregates accumulate within lysosomes, even in the absence of CSPα[−/−]-driven lysosomal pathology. Thus, we tested whether aged Tg-αSyn^A53T mice, which display severe αSyn-driven pathology, have αSyn aggregates within their lysosomes.

We found pathogenic αSyn species, but not monomeric αSyn, in dextranosomes (heavy lysosomes loaded with dextran to enhance iso-lation-purity) isolated from the brains of 6-month-old Tg-αSyn^A53T mice (Fig. 1). These lysosomal fractions precisely coincided with pathogenic αSyn species and had highly enriched lysosomal proteins as well as cathepsin-D activity, with negligible contamination from other orga-nelle protein-markers or mitochondrial and peroxisomal enzyme activities (Fig. 1). This experiment indicates that isolated intact lyso-somes from aged Tg-αSyn^A53T mice contain pathogenic αSyn species.

### Pathogenic species of αSyn accumulate within the lumen of neuronal lysosomes

The lysosomes studied above are a mixture of lysosomes from all brain cells. To isolate and study lysosomes specifically from neurons, we generated transgenic mice expressing ^HALAMP1^Myc—late-endosomal/lysosomal protein LAMP1 epitope-tagged on its luminal domain with tandem 2xHA, and on its cytoplasmic tail with 2xmyc—driven by the neuron-specific synapsin-I promoter (Fig. 2a−d and Supplementary Fig. S2a−d), enabling affinity-isolation of neuronal late-endosomes/lysosomes from mouse brains. The Tg-^HALAMP1^Myc mice were then crossed to the Tg-αSyn^A53T mice to generate Tg-αSyn^A53T/Tg-^HALAMP1^Myc progeny (Supplementary Fig. S3a−g). Immunoisolation experiments from aged Tg-αSyn^A53T/Tg-^HALAMP1^Myc mouse brains indicated that pathogenic versions of αSyn co-isolate with neuronal late-endosomes/lysosomes (Fig. 3a). We found that ~10% of the SDS-resistant αSyn aggregates, and ~20% of the more mature αSyn aggregates (fila-mentous and amyloid-type) reside within neuronal late-endosomes/lysosomes (Fig. 3a). No monomeric αSyn resided in neuronal late-endosomes/lysosomes (Fig. 3a). Together with the dextranosome-isolation experiments (Fig. 1), this result indicates that a non-trivial portion of pathogenic αSyn species in the brain is found within neu-ronal lysosomes.

Electron microscopy showed at the ultrastructural level that αSyn aggregates were localized within the lysosomal lumen of cortical neurons in 6-month-old Tg-αSyn^A53T mouse brains (Fig. 3b). This was further corroborated by limited proteolysis of immunoisolated lyso-somes from 6-month-old αSyn^A53T/Tg-^HALAMP1^Myc mice, where αSyn aggregates were resistant to proteinase K digestion, only in the absence of a detergent, similar to the lysosome-luminal enzyme cathepsin-L, and in contrast to the lysosomal vATPase subunits that are exposed on the cytosolic side of lysosomal membrane (Fig. 3c). Alto-gether, these experiments localize αSyn aggregates inside the lysoso-mal lumen.

To understand the fate of these lysosomal αSyn aggregates, we developed a neuronal model which accumulates pathogenic αSyn species. Compared to Tg-αSyn^A53T mice, homozygous Tg-αSyn^A53T mice (Tg^x2-αSyn^A53T), which express twice the amount of transgenic αSyn^A53T, have shorter lifespans, earlier onset of neuromuscular pathology, and begin accumulating pathogenic αSyn species earlier—by only 6 weeks of age (Supplementary Fig. S4a−f). Accordingly, long-lived primary neurons from Tg^x2-αSyn^A53T mice also accrued pathogenic αSyn species by 6 weeks, including aggregates detectable by antibodies against αSyn fibrils and amyloid-type conformations (Supplementary Fig. S5a, b), which were Triton X-100 insoluble, extractable in SDS, and dis-rupted by urea (Supplementary Fig. S5c).

Importantly, we further verified the accumulation of these pathogenic αSyn aggregates in lysosomes using a second independent immunoisolation approach. Isolation of lysosomes from Tg^x2-αSyn^A53T neurons using the HA-tagged lysosomal protein TMEM192 (previously characterized in[36,37]) captured αSyn aggregates within neuronal lyso-somes at DIV49, but not earlier at DIV35 and DIV21 (Supplementary Fig. S6). No signal for these pathological αSyn species was detected in the lysosomes isolated from DIV49 αSyn knockout (αSyn[−/−]) neurons.

We then applied two independent proximity-labeling techniques to specifically mark and track the pathogenic αSyn species located within neuronal lysosomes. First, we targeted Apex-2 to the lysosomal lumens of Tg^x2-αSyn^A53T primary neurons. We lentivirally expressed a synapsin-1 promoter-driven construct, comprised of Apex-2 fused to the luminal domain of a truncated version of LAMP1 (^Apex-2LAMP1) (Supplementary Fig. S7a−c), which proximity-labels the luminal con-tents of neuronal late-endosomes/lysosomes with biotin. In these Tg^x2-αSyn^A53T neurons expressing ^Apex-2LAMP1, we found that pathogenic αSyn species as well as lysosomal proteins (typified by cathepsin-L) were biotinylated (Fig. 3d). Second, we used an in situ proximity liga-tion assay, and found that filamentous αSyn aggregates co-localized with cathepsin-D, i.e., within the lysosomal lumen of Tg^x2-αSyn^A53T primary neurons (Fig. 3e). Together, these results show that patho-genic αSyn species accumulate within the lumen of neuronal lysosomes.

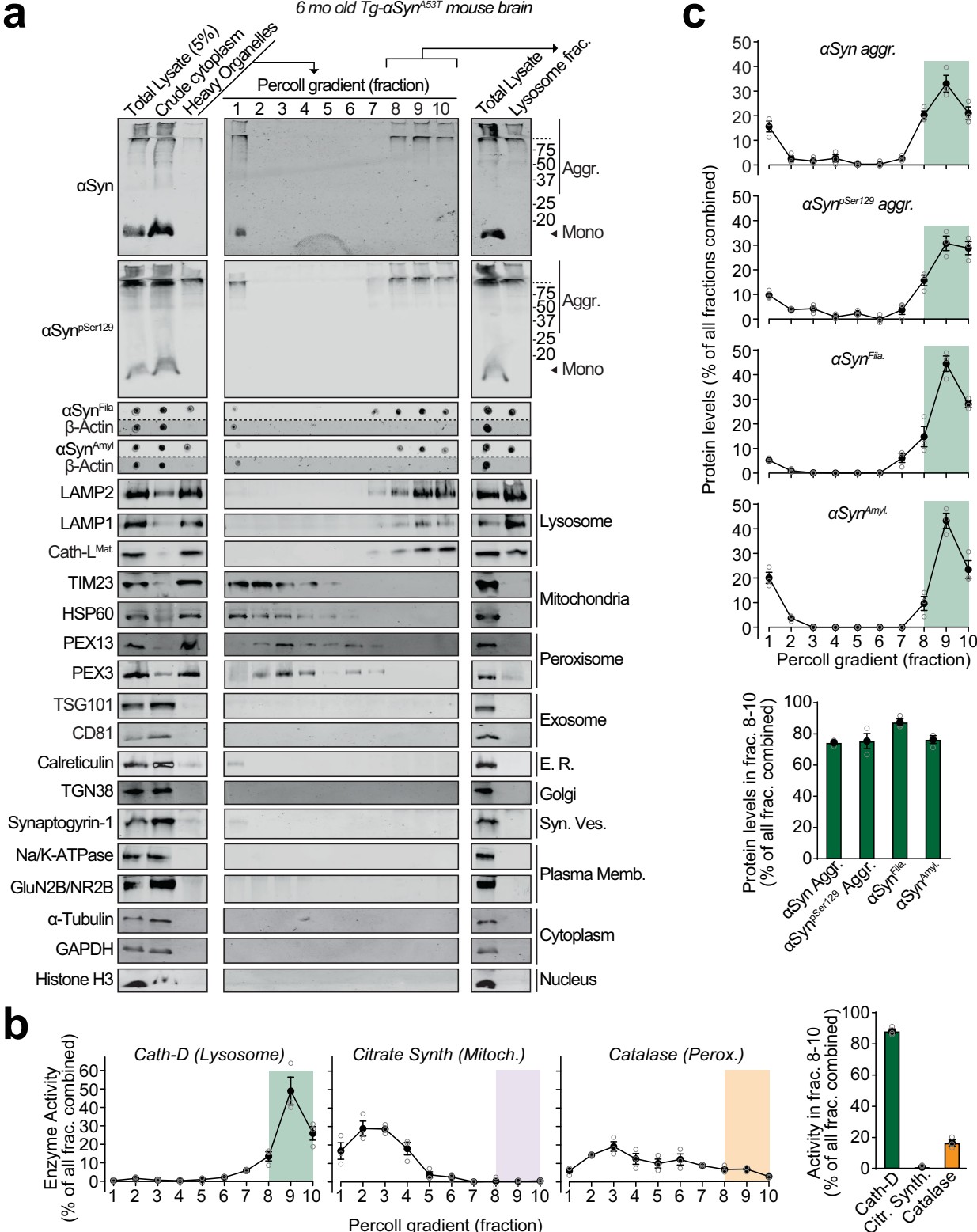

**Fig. 1 | Pathogenic αSyn aggregates accumulate within lysosomes of aged Tg-αSyn^A53T mice.** Lysosomes were isolated from 6-month-old Tg-αSyn^A53T mouse brains via Percoll gradient centrifugation of the heavy organelle fraction—which contained peroxisomes, heavy lysosomes loaded in vivo with dextran-70, and mitochondria swollen ex vivo by CaCl₂. **a** Lysosome (dextranosome) enrichment was determined by immunoblotting for markers of indicated organelles, compared to the respective levels in the total lysate input. Membrane-matched dot blots (β-Actin = loading control) are separated by dashed lines. **b** Left panels—Activities of enzymes contained within lysosomes (cathepsin-D), mitochondria (citrate

synthase), and peroxisomes (catalase) were measured, testing for isolation of intact organelles. Right panel—Summary graph of enzyme activity present in the combined "lysosomal fractions" (fractions 8–10). **c** Top panels—Levels in each Percoll gradient fraction of pathogenic αSyn species: Aggregated (αSyn Aggr), aggregates phosphorylated at Ser129 (αSyn^pSer129 Aggr), filamentous (αSyn^Fila), and amyloid-type (αSyn^Amyl). Bottom panel—Summary graph of these αSyn species present in the combined "lysosomal fractions" (fractions 8–10). (n = 3). All data are shown as means ± SEM, where "n" represents mouse brains.

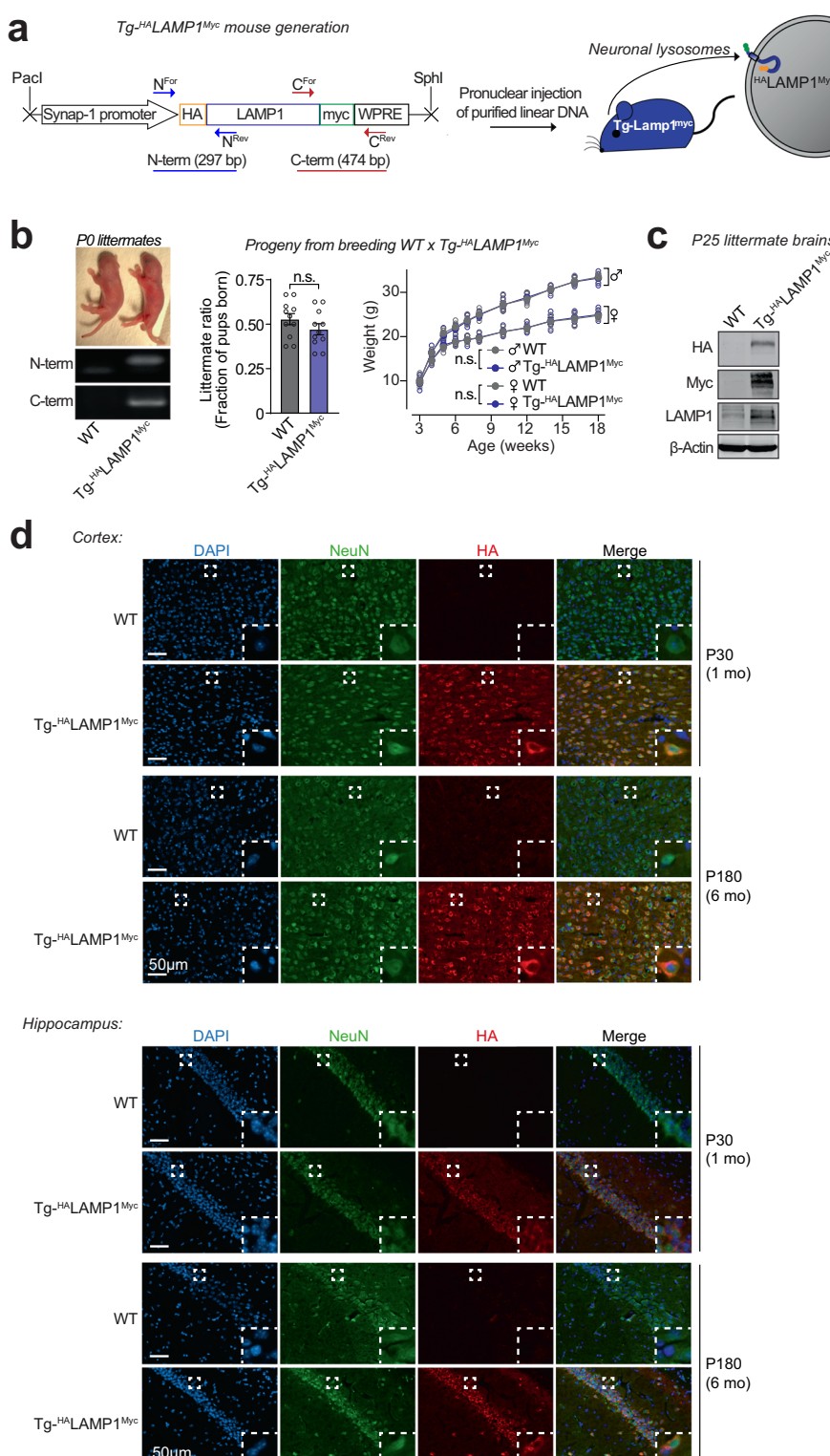

**Fig. 2 | Development of a transgenic mouse model (NeuLyso-Tag mouse) to isolate neuronal lysosomes from brains. a** A lentiviral vector expressing HALAMP1Myc via the neuron-specific synapsin-1 promoter was used to generate Tg-HALAMP1Myc (or "NeuLyso-Tag") mice via pronuclear injection of linearized DNA comprising the synapsin-1 promoter and the HALAMP1Myc cDNA. **b** Tg-HALAMP1Myc mice were PCR-genotyped using primer-pairs at the N- and the C-terminal ends of the HALAMP1Myc cDNA. Single insertion-site for the transgene was suggested by equal numbers of WT and Tg- HALAMP1Myc pups born per litter (*n* = 11 litters). There was also no effect of the transgene on birth-ratio and growth (indicated by weight gain; *n* = 5 per group) of the Tg- HALAMP1Myc mice compared to WT mice. **c** Immunoblots show that HALAMP1Myc protein is detectable in mouse brains by

immunoblots against the N-terminal HA tag and the C-terminal myc tag (representative of *n* = 7 littermates of each genotype). **d** Histological colocalization of anti-HA immunofluorescence with NeuN staining in the cortex and hippocampus shows that HALAMP1Myc construct is expressed in the neurons of Tg-HALAMP1Myc mouse brains at 1 and 6 months of age (representative of >10 stained sections from *n* = 3 littermates of each genotype). All data represent means ± SEM. n.s. not significant, by paired 2-tailed Student's *t* test for littermate ratio and by RM 2-way ANOVA for weight gain over age (each mouse matched over age). *Note:* The FSW-HALAMP1Myc lentiviral vector is described and tested in Supplementary Fig. S2, and further characterization of the Tg- HALAMP1Myc mice is included in Supplementary Fig. S3.

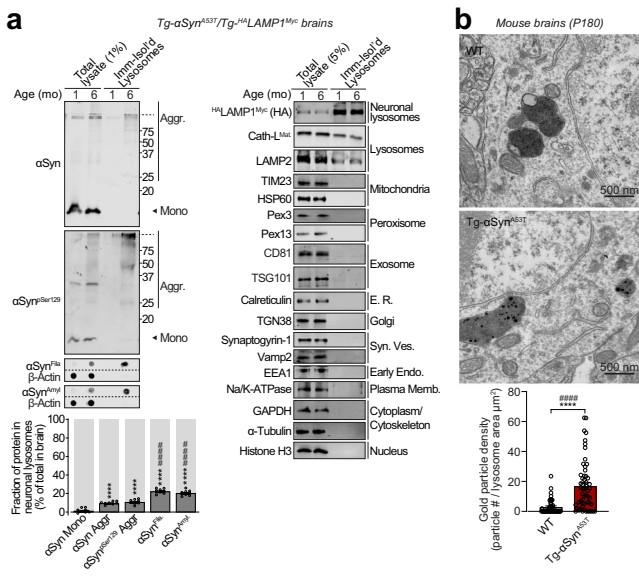

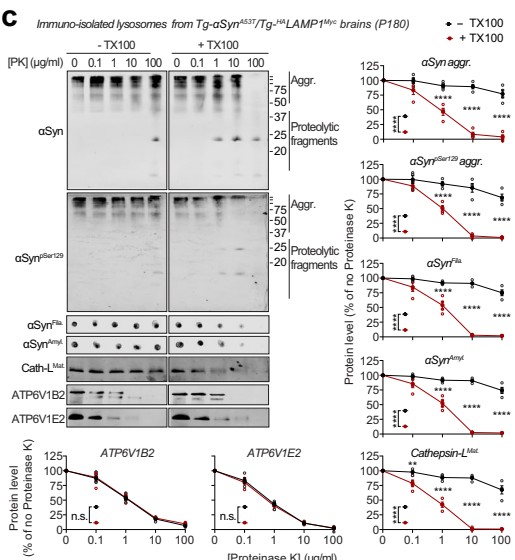

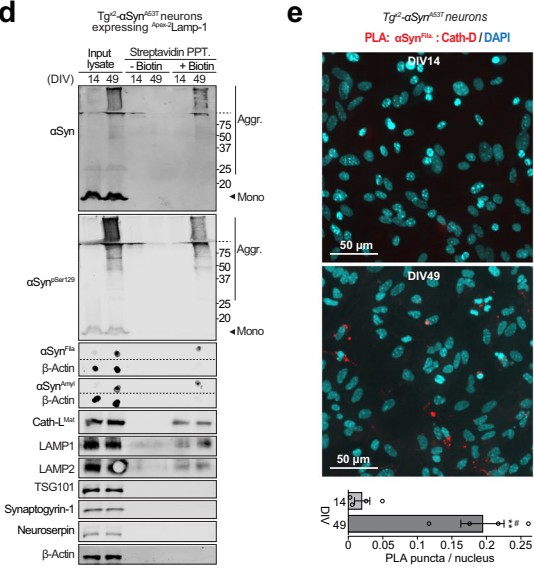

**Fig. 3 | Pathogenic αSyn species accumulate within neuronal lysosomes in mouse brains and primary neurons. a** Neuronal late-endosomes/lysosomes immunoisolated from the brains of 1- and 6-month-old Tg-αSyn$^{A53T}$/Tg-$^{HA}$LAMP1$^{Myc}$ mice using anti-myc antibodies were analyzed by immunoblotting for levels of αSyn, αSyn$^{pSer129}$, αSyn$^{Fila}$, and αSyn$^{Amyl}$ (Mono monomer, Aggr aggregates). The fraction of these αSyn species residing within neuronal lysosomes at 6 months of age (bottom graph) was back-calculated from the fraction of each αSyn species immunocaptured (αSyn species immunocaptured/total input), normalized to the fraction of neuronal late-endosomes/lysosomes captured—indicated by the fraction of $^{HA}$LAMP1$^{Myc}$ captured (HA blot immunocaptured/total input). Membrane-matched dot blots (β-Actin = loading control) are separated by dashed lines, and immunoblots against markers of lysosomes and of other organelles are also shown (n = 8). **b** Electron micrographs showing increased αSyn aggregate-labeling within the lysosomal lumen in 6-month-old Tg-αSyn$^{A53T}$ brains (2 lysosomes shown with 12 and 5 gold particles) compared to WT brains (2 lysosomes shown with 1 possible and 0 gold particles). (n = 50 lysosomes for WT and n = 55 lysosomes for Tg-αSyn$^{A53T}$). **c** Lysosomes isolated from the brains of 6-month-old Tg-αSyn$^{A53T}$/$^{HA}$LAMP1$^{Myc}$ mice (as in **a**) were subjected to on-bead limited proteolysis using the indicated concentrations of proteinase K (PK) in the absence or presence of 0.1% Triton X-100. Immunoblots for αSyn (αSyn, αSyn$^{pSer129}$, αSyn$^{Fila}$, and αSyn$^{Amyl}$), mature cathepsin-L (Cath-L$^{Mat}$; lysosome lumen), and vATPase subunits ATP6V1B2 and ATP6V1E2 (lysosomal proteins exposed on the cytosolic side) (n = 4 brains, isolation, and proteolysis). **d** Tg$^{x2}$-αSyn$^{A53T}$ primary cultures lentivirally expressing a synapsin-1 promoter-driven $^{Apex-2}$LAMP1 construct were subjected to proximity-labeling with biotin at 14 or 49 days in vitro (DIV). Labeled proteins were pre-cipitated on streptavidin-coated magnetic beads and immunoblotted for the indi-cated pathogenic αSyn species, as well as for the marker proteins cathepsin-L (lysosomes), LAMP1 and LAMP2 (late-endosomes/lysosomes), TSG101 (exosomes), synaptogyrin-1 (synaptic vesicles), neuroserpin (non-lysosomal/constitutively secreted neuronal protein), and β-actin (cytosol). Membrane-matched dot blots (β-Actin = loading control) are separated by dashed lines. (representative of n = 3). **e** Tg$^{x2}$-αSyn$^{A53T}$ primary cultures were subjected to in situ proximity ligation assay (PLA) at either DIV14 (top) or at DIV49 (bottom), with primary antibodies against α-Syn$^{Fila}$ and cathepsin-D (Cath-D). Quantification of PLA puncta per nucleus is shown, performed identically between images from DIV14 and 49 (n = 4 cultures and PLA experiments; data from 3 images averaged for each n). Data represent means ± SEM. Each "n" corresponds to independently aged mouse littermates in **a**, lysosomes in **b**, independent mouse brains, immunoisolation and proteolysis experiments in **c**, independent primary neuron cultures in **d**, and independent pups, cultures, and PLA experiments in **e**. ****P < 0.0001 by 1-way ANOVA with Dunnett multiple-comparison correction and ####P < 0.0001 by non-parametric Kruskal−Wallis test with Dunn's multiple-comparison adjustment; all tests com-paring levels of each αSyn aggregate-type to αSyn monomer in **a**; ****P < 0.0001 by unpaired 2-tailed Student's t test and ####P < 0.0001 by non-parametric Mann−Whitney test in **b**; 2-way ANOVA with Bonferroni multiple comparisons post-test (compared at each PK concentration) in **c**; and **P < 0.01 by unpaired 2-tailed Student's t test and #P < 0.05 by non-parametric Mann−Whitney test in **e**.

## Pathogenic αSyn species are secreted from neurons via SNARE-dependent lysosomal exocytosis

Release of αSyn from neurons, particularly the seeding-competent pathogenic versions of αSyn, is considered a key step in the spatial progression of synucleinopathies[38], and αSyn has been documented in extracellular fluids of PD patients[6]. We also found pathogenic species of αSyn in the cerebrospinal fluid of aged Tg-αSyn$^{A53T}$ mice (Fig. 4a). Importantly, this extracellular pool of αSyn aggregates is not envel-oped in membranes, as they were not protected from proteinase K digestion (Fig. 4a), and their proteolytic susceptibility was unaffected by the presence of detergent, akin to the lysosomal luminal protein cathepsin-L and neuroserpin—a non-lysosomal secreted protein (Fig. 4a). This was contrary to the well-protected contents of exo-somes, typified by TSG101, which becomes susceptible to proteolysis only in the presence of detergent (Fig. 4a).

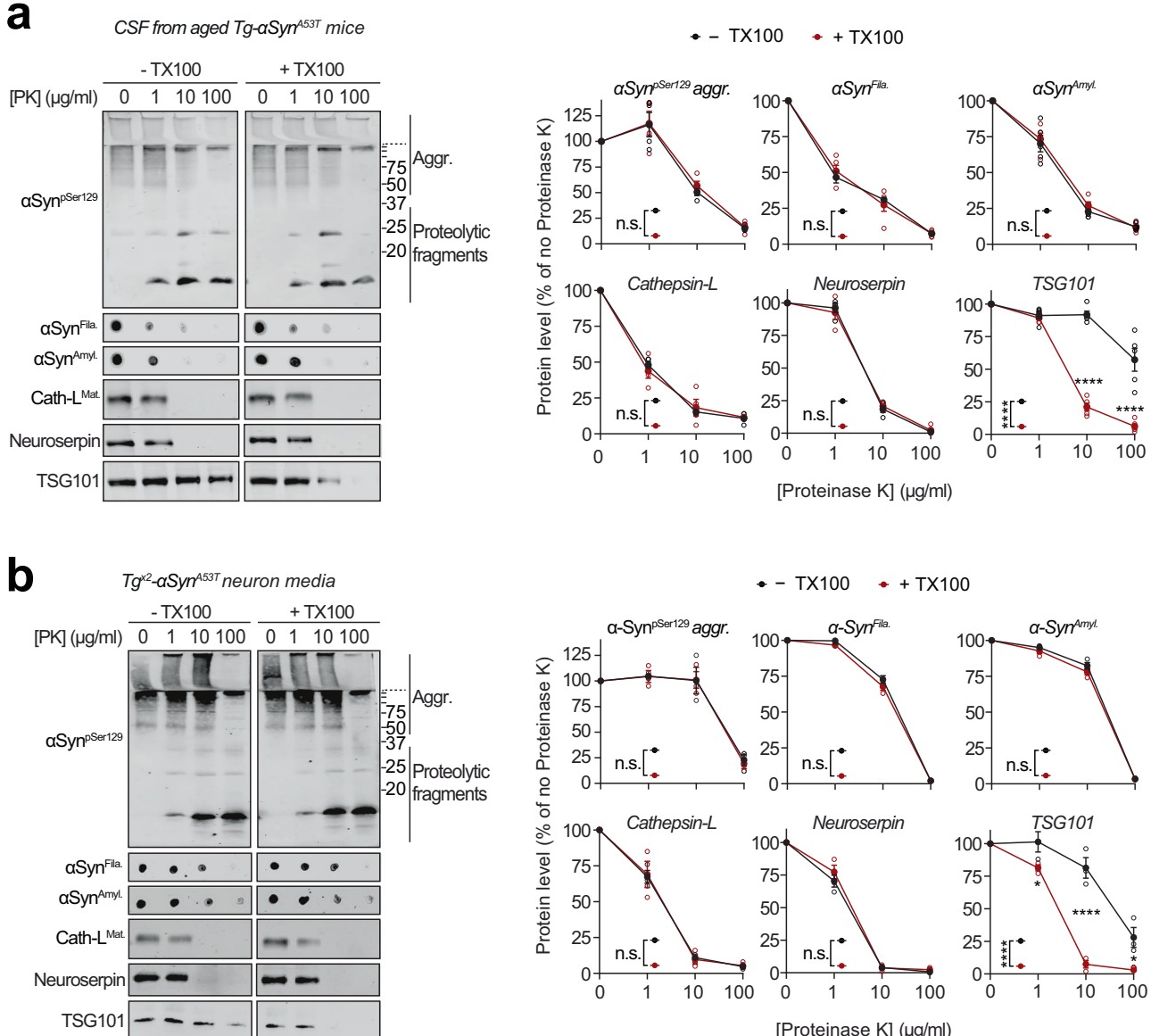

**Fig. 4 | Extracellular αSyn species in cerebrospinal fluid and in primary neuron media are not membrane-enveloped. a** Cerebrospinal fluid (CSF) collected from 6-month-old Tg-αSyn$^{A53T}$ mice were subjected to limited proteolysis using indicated concentrations of proteinase K (PK) in the absence or presence of 0.1% Triton X-100. Immunoblots for αSyn protein levels (αSyn$^{pSer129}$, αSyn$^{Amyl}$, and αSyn$^{Fila}$), mature cathepsin-L (Cath-L$^{Mat}$), TSG101, and neuroserpin ($n = 5$ for αSyn$^{Amyl}$, αSyn$^{Fila}$,

TSG101; $n = 4$ for αSyn$^{pSer129}$, Cath-L$^{Mat}$). **b** Media collected from Tg$^{x2}$-αSyn$^{A53T}$ neuron cultures at DIV49 was subjected to limited proteolysis in presence or absence of 0.1% Triton X-100 and analyzed as in **a** ($n = 3$). All data represent means ± SEM. Each 'n' is an independent collection (CSF or medium) and proteolysis experiment. n.s. not significant; *$P < 0.05$; ****$P < 0.0001$ by 2-way ANOVA with Bonferroni multiple comparisons post-test (compared at each PK concentration).

The pathogenic αSyn species were also detected in the medium of Tg$^{x2}$-αSyn$^{A53T}$ neurons collected at DIV49, but were not yet present at DIV35 or in the media of DIV49 αSyn$^{-/-}$ and WT neurons (Supplementary Fig. S5d). This extracellular pool of αSyn aggregates, similar to that in cerebrospinal fluid, was not protected from proteinase K digestion (Fig. 4b), indicating that it is not membrane-enveloped. Additionally, proteolytic susceptibility of these αSyn aggregates, as well as that of cathepsin-L and neuroserpin were unaffected by the presence of detergent (Fig. 4b). In contrast, membrane-enveloped exosomal contents were well-protected, typified by TSG101, which became susceptible to proteolysis only in the presence of detergent (Fig. 4b). Notably, we found no evidence of neuronal death as a potential source of extracellular αSyn aggregates (Supplementary Fig. S5e, f).

Accumulation of the pathogenic αSyn species inside lysosomes, together with their un-enveloped presence in the extracellular space,

prompted us to investigate whether lysosomes are directly releasing the αSyn aggregates contained inside them.

We thus collected the medium from Tg$^{x2}$-αSyn$^{A53T}$ primary neurons, in which the lysosomal contents had been biotinylated via neuron-specific expression of $^{Apex-2}$LAMP1 (as in Fig. 3b and Supplementary Fig. S7). We found that lysosomal contents, typified by cathepsin-L, as well as pathogenic αSyn species, could be precipitated from the medium of Tg$^{x2}$-αSyn$^{A53T}$ neurons via their biotin-modification (Fig. 5a). In contrast, exosomal contents, typified by TSG101, were not biotinylated/precipitated. This result points to lysosomal exocytosis−SNARE-dependent fusion of lysosomes with the plasma membrane−as the likely mechanism for the exit of αSyn aggregates from neurons.

SNARE-dependent lysosomal exocytosis is triggered by cytosolic calcium[39], which in turn is modulated by neuronal activity and/or

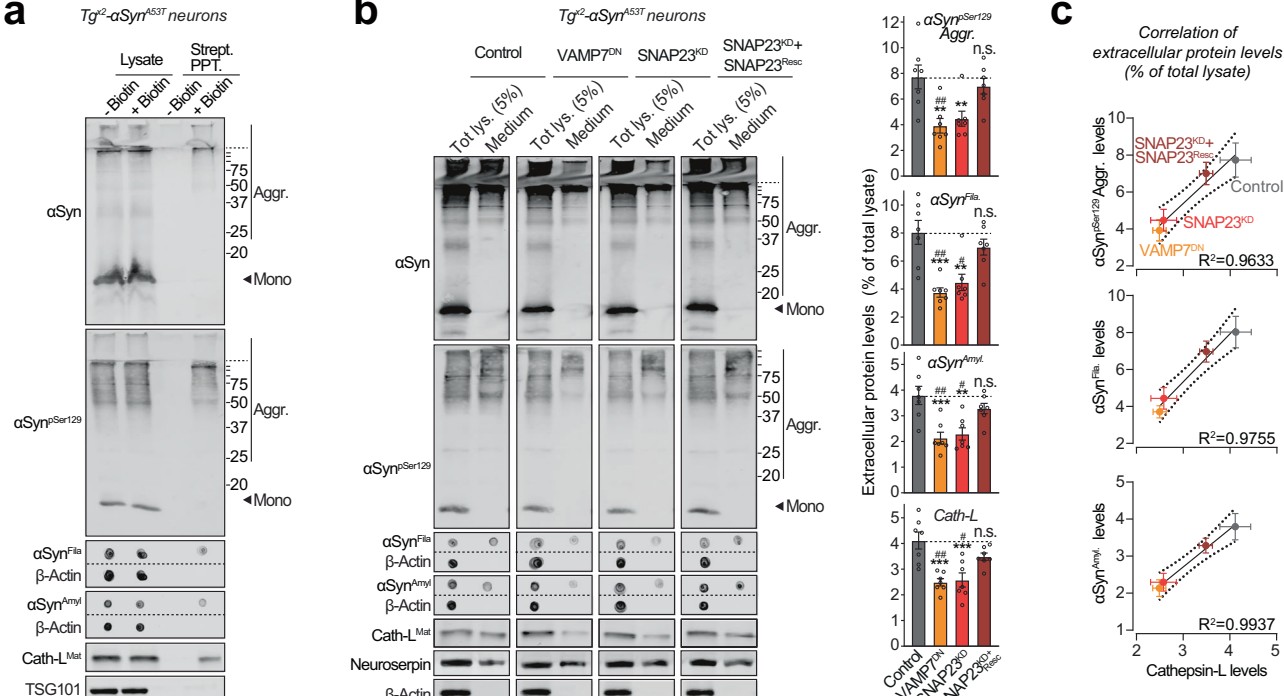

**Fig. 5 | Pathogenic αSyn species are secreted from neurons via SNARE-dependent lysosomal exocytosis. a** Tg$^{x2}$-αSyn$^{A53T}$ primary neurons lentivirally expressing a synapsin-1 promoter-driven $^{Apex-2}$LAMP1 (infected on DIV7) were subjected to proximity-labeling on DIV 47. Biotinylated proteins released into the media during 48 h (by DIV49) were precipitated on streptavidin-magnetic beads and immunoblotted for the indicated αSyn species (αSyn, αSyn$^{pSer129}$, αSyn$^{Amyl}$, and αSyn$^{Fila}$), lysosome-luminal protein cathepsin-L, and exosome luminal protein TSG101. Strept. PPT. streptavidin precipitate. Membrane-matched dot blots (β-Actin = loading control) are separated by dashed lines. (representative of $n = 3$). **b** Media collected over 2 days (DIV 47–49) from Tg$^{x2}$-αSyn$^{A53T}$ primary neurons lentivirally expressing control (GFP), GFP-VAMP7$^{DN}$ fragment, SNAP23$^{KD}$ shRNA, and SNAP23$^{KD}$ shRNA plus knockdown-resistant SNAP23 rescue-construct (infected on DIV7) were immunoblotted for the indicated αSyn species (αSyn, αSyn$^{pSer129}$, αSyn$^{Amyl}$, and αSyn$^{Fila}$), cathepsin-L (Cath-L$^{Mat}$; late-endosome/lysosome-luminal), neuroserpin (non-lysosomally/constitutively secreted from neurons), and β-actin

(cytoplasmic). Membrane-matched dot blots (β-Actin = loading control) are separated by dashed lines. For quantification (graphs on the right), protein level in the medium was normalized to its levels in total cellular lysate (5% loaded) ($n = 7$). **c** Correlation between the levels of pathogenic αSyn species and the levels of cathepsin-L released in the medium, upon the indicated manipulation of SNARE-dependent lysosomal exocytosis; derived from data shown above in **b**. Linear regression is shown with dotted lines depicting the 95% confidence intervals, and Pearson's correlation coefficient is indicated on the bottom right of each graph. ($n = 7$). All data represent means ± SEM. Each "$n$" corresponds to a separate mouse litter used for a batch of neuronal culture and infection. In **b** n.s. not significant; **$P < 0.01$; ***$P < 0.001$ by 1-way ANOVA with Dunnett multiple-comparison correction compared to control; and #$P < 0.05$; ##$P < 0.01$; by non-parametric Kruskal–Wallis test with Dunn's multiple-comparison adjustment compared to control.

exchange with organellar calcium stores. We first pharmacologically modulated neuronal activity in Tg$^{x2}$-αSyn$^{A53T}$ neurons, by using TTX or CNQX/APV to inhibit activity, and K$^+$ or bicuculline methiodide to enhance activity. Release of αSyn aggregates as well as that of cathepsin-L directly correlated with neuronal activity (Supplementary Fig. S8a). Next, we inhibited the release of calcium from the endoplasmic reticulum using dantrolene/2-APB, or inhibited the influx of extracellular calcium using YM-58483, in Tg$^{x2}$-αSyn$^{A53T}$ neurons (Supplementary Fig. S8b). Both strategies reduced the release of αSyn aggregates and cathepsin-L into the medium (Supplementary Fig. S8b). Neither the manipulation of neuronal activity nor the reduction of cytoplasmic calcium caused neuron death (Supplementary Fig. S8c, d). These pharmacological experiments further suggest that release of pathogenic αSyn species from neurons is mediated via calcium-regulated SNARE-dependent lysosomal exocytosis.

To test this hypothesis directly, we modulated the SNARE proteins responsible for lysosome-to-plasma membrane fusion. First, we identified VAMP7 as the prominent v-SNARE in immunoisolated lysosomes (Supplementary Fig. S9a). We then screened for the Qb-type t-SNAREs which interact and thus co-immunoprecipitate with VAMP7. SNAP23 was the clearest Qb/t-SNARE candidate (Supplementary Fig. S9a).

Based on these results and prior studies validating them[40], we tested four strategies to disrupt lysosomal exocytosis: shRNA knockdown of either SNAP23 (SNAP23$^{KD}$) or VAMP7 (VAMP7$^{KD}$), and overexpression of dominant-negative fragments of either SNAP23 (SNAP23$^{DN}$) or VAMP7 (VAMP7$^{DN}$). In wild-type primary neurons, the release of cathepsin-L was used to signal lysosomal exocytosis, opposed to neuroserpin—a non-lysosomal secreted protein. VAMP7$^{KD}$ and SNAP23$^{DN}$ had no significant effect on lysosomal exocytosis, but VAMP7$^{DN}$ severely reduced it by ~80% and SNAP23$^{KD}$ partially reduced it by ~40%, with no effect on cell viability (Supplementary Fig. S9b–d). Therefore, we applied the latter two strategies to disrupt lysosomal exocytosis next.

Importantly, in Tg$^{x2}$-αSyn$^{A53T}$ neurons, both VAMP7$^{DN}$ and SNAP23$^{KD}$ strategies reduced the release of pathogenic αSyn species into the medium, and the effect of SNAP23$^{KD}$ was rescued by overexpression of a knockdown-resistant version of wild-type SNAP23 (Fig. 5b). Moreover, the release of pathogenic αSyn species was significantly correlated with the change in lysosomal exocytosis, measured as cathepsin-L released into the medium (Fig. 5b, c). No β-actin was detectable in the medium (Fig. 5b). These results suggest that SNARE-dependent exocytosis of lysosomes is essential and rate-determining for the release of pathogenic αSyn species from neurons.

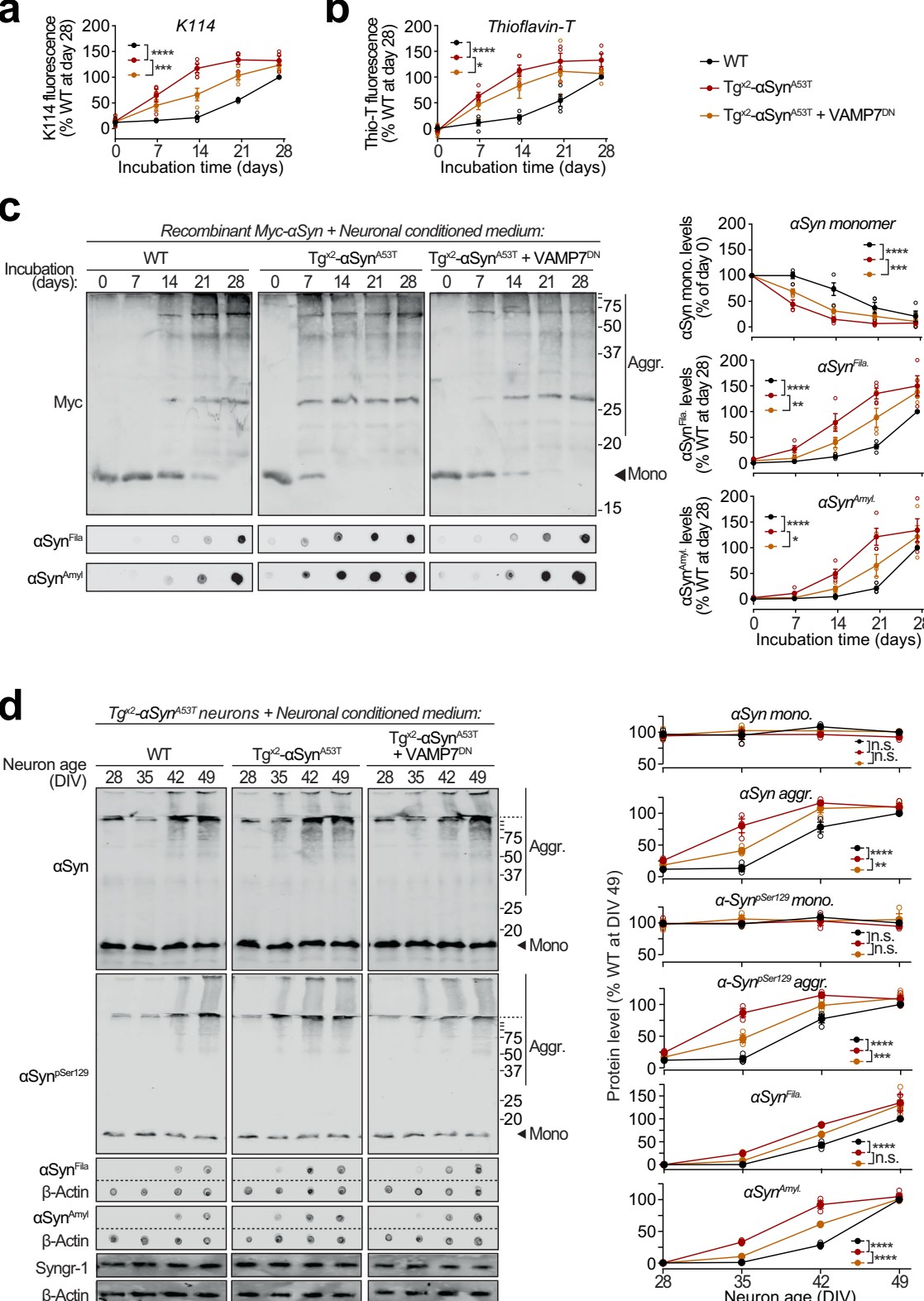

**αSyn species released via lysosomal exocytosis can seed aggregation of purified recombinant αSyn and of endogenous αSyn in neurons**

Pathogenic spread requires that the released αSyn is able to template or "seed" the assembly of monomeric αSyn into amyloid-type aggregates.

To test whether the pathogenic αSyn species exocytosed from neurons are seeding-competent, we performed in vitro aggregation of purified recombinant myc-tagged wild-type αSyn (Supplementary Fig. S10a), in the presence of extracellular media collected from neuron cultures generated either from wild-type mice, or from Tg$^{x2}$-αSyn$^{A53T}$ mice with or without lentiviral expression of VAMP7$^{DN}$ fragment (Fig. 6a–c). Aggregation kinetics, measured by amyloid-binding fluorescent dye K114 (Fig. 6a) or Thioflavin-T (Fig. 6b), were enhanced by medium from Tg$^{x2}$-αSyn$^{A53T}$ neurons, compared to the medium from

**Fig. 6 | Seeding of aggregation of recombinant and neuronally expressed αSyn by pathogenic αSyn species exocytosed from neurons. a–c** Recombinant purified myc-tagged αSyn (myc-αSyn) was shaken at 37 °C in presence of concentrated extracellular medium from mouse cortical neuron cultures collected over 48 h (DIV 47–49), generated either from wild-type (WT) mice, or from Tg^{x2}-αSyn^{A53T} mice with or without lentiviral expression of VAMP7 dominant-negative fragment (VAMP7^{DN}; infected at DIV7). Aggregation of myc-αSyn was analyzed at the indicated days of incubation by the following assays: **a** Congo-red derivative, amyloid-binding dye K114 fluorescence at 390/535 nm ($n = 4$). **b** Amyloid-binding dye Thioflavin-T fluorescence at 450/485 nm ($n = 4$). **c** Quantitative immunoblotting for the myc epitope-tag, where aggregation is measured as disappearance of monomeric myc-αSyn (top; $n = 4$); dot-blotting for filamentous myc-αSyn aggregates using αSyn^{Fila}

antibody (middle; $n = 4$); and dot-blotting for amyloid-type myc-αSyn aggregates using αSyn^{Amyl} A11 antibody (bottom; $n = 4$). **d** Concentrated media collected over 48 h (DIV 47–49) from WT neurons, or from Tg^{x2}-αSyn^{A53T} neurons with or without lentiviral expression of VAMP7^{DN} was added to the medium of Tg^{x2}-αSyn^{A53T} neurons at DIV14 and removed 48 h later by replacing the medium. Neurons were harvested at indicated times, and αSyn aggregation was measured by immunoblotting, and normalized to synaptogyrin-1 (Syngr-1) levels. Membrane-matched dot blots (β-Actin = loading control) are separated by dashed lines. ($n = 3$). All data represent means ± SEM, where each "n" is an independent media collection and aggregation experiment. n.s. not significant; *$P < 0.05$; **$P < 0.01$; ***$P < 0.001$; ****$P < 0.0001$ by 2-way ANOVA.

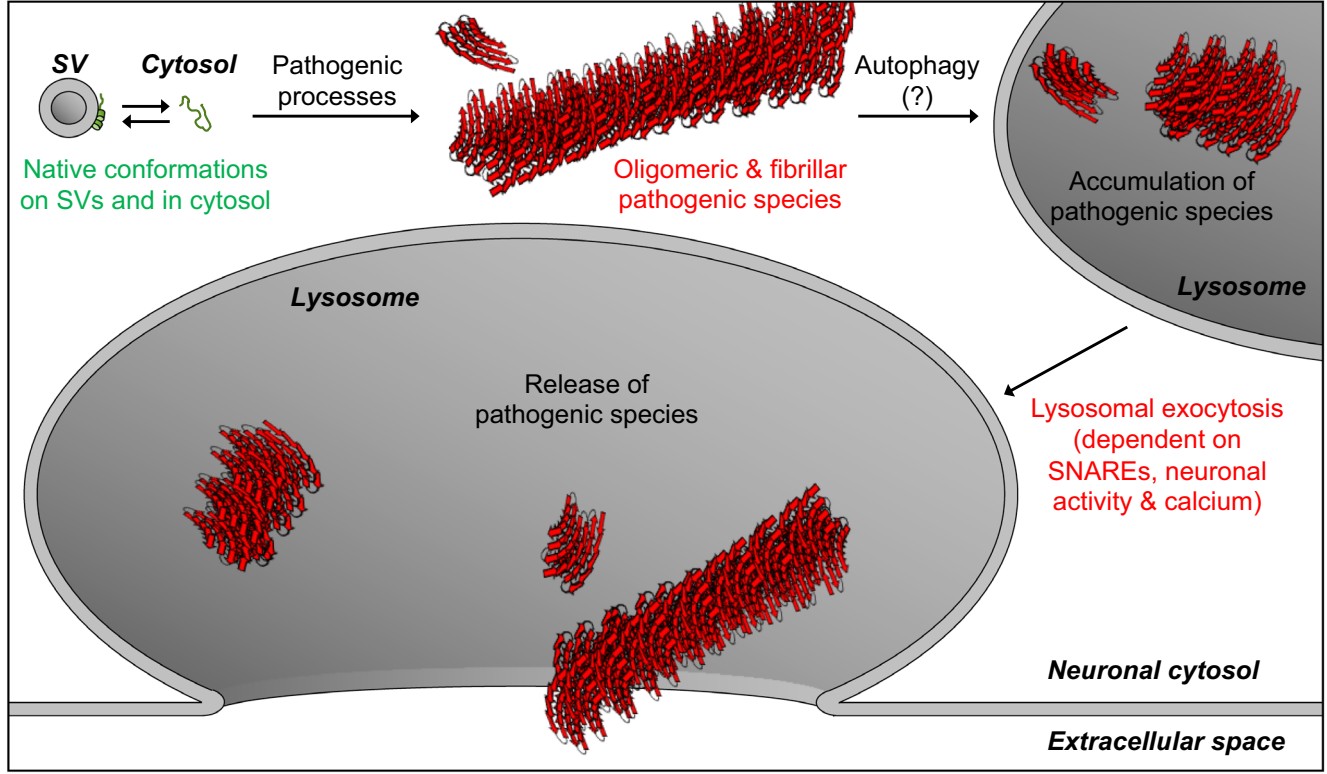

**Fig. 7 | Model of our findings: Lysosomal exocytosis releases pathogenic αSyn species from neurons.** Native αSyn is expressed as a natively unstructured monomer in the cytosol, in equilibrium with an α-helical conformer upon binding the highly curved synaptic vesicle (SV) membrane. Pathogenic processes oligomerize αSyn into β-sheet-rich amyloid-type aggregates of various sizes. We demonstrate here that these larger pathogenic species, generated in the neuronal cytosol, accumulate in neuronal lysosomes over time, possibly due to their high structural and metabolic stability. SNARE-dependent lysosomal exocytosis releases these non-membrane-enveloped aggregates, and this release is upregulated by neuronal activity and cytosolic Ca²⁺.

wild-type neurons. Suppressing lysosomal exocytosis in the Tg^{x2}-αSyn^{A53T} neurons, via VAMP7^{DN} expression, diminished the aggregation-promoting effect of the Tg^{x2}-αSyn^{A53T} medium (Fig. 6a, b). When myc-αSyn aggregation kinetics were measured by quantitative immunoblotting for either disappearance of myc-tagged αSyn (myc-αSyn) monomers or for the appearance of filamentous and amyloid-type aggregates (shown in αSyn^{Fila} and αSyn^{Amyl} blots), we again found that compared to the wild-type neuron medium, aggregation was augmented by the Tg^{x2}-αSyn^{A53T} medium, and the expression of VAMP7^{DN} fragment diminished this effect (Fig. 6c). Notably, the effect of wild-type neuron medium on myc-αSyn aggregation rate was indistinguishable from the effect of unconditioned medium and αSyn^{−/−} neuron medium (Supplementary Fig. S10b). In all the assays, a lag-time of nearly 2 weeks is apparent before the appearance of amyloid signal with the wild-type medium (Fig. 6a–c). Importantly, this lag is shortened or eliminated by the secreted aggregates in Tg^{x2}-αSyn^{A53T} medium (Fig. 6a–c), with a reduced effect when lysosomal exocytosis is suppressed by VAMP7^{DN} fragment (Fig. 6a–c).

Moreover, in Tg^{x2}-αSyn^{A53T} primary neurons, appearance of αSyn aggregates was accelerated by media from Tg^{x2}-αSyn^{A53T} neurons, but not from wild-type neuron media (Fig. 6d). Importantly, media from Tg^{x2}-αSyn^{A53T} neurons expressing the VAMP7^{DN} fragment—which inhibits lysosomal exocytosis, had a significant reduction in this effect (Fig. 6d).

These results substantiate that αSyn species exocytosed by lysosomes are capable of amyloid-nucleation via permissive templating, as well as spreading this seeding activity into live neurons.

## Discussion

Release of αSyn aggregates from neurons is thought to transmit neuropathology to other yet-unaffected neurons. Here, we find that pathogenic αSyn species accumulate within neuronal lysosomes in mouse brains and in primary neurons, and are then released from neurons via SNARE-dependent lysosomal exocytosis (Fig. 7).

It is important to note that for αSyn pathology to transmit from neuron-to-neuron, the lysosomal exocytosis mechanism explains the

following observations, which underscore that the extracellular pool of pathogenic αSyn is likely not membrane-enveloped: (a) the stark pathology and rapid propagation caused by inoculated pre-formed fibrils[10,11,41–44], (b) the efficacy of humoral and cellular immunotherapy against αSyn (reviewed in ref. [45]), (c) the efficacy of passive immunization using antibodies against αSyn epitopes (reviewed in ref. [45]), and (d) much of the extracellular αSyn has been found as non-enveloped[6,46,47], while a minority of αSyn is contained in extracellular vesicles or exosomes[12,18,20]. However, other proposed means of αSyn aggregate-transmission might simultaneously contribute to pathological propagation in synucleinopathies[38].

Key gaps still remain in how the pathogenic αSyn species are targeted and trafficked to lysosomes, as well as in the fate of extracellular αSyn aggregates following their release from neurons:

The aggregates/fibrils of αSyn could form inside the acidified lumen of lysosomes[48], e.g., from monomers taken up via chaperone-mediated autophagy or macroautophagy of synaptic vesicles, especially if their degradation is delayed due to reduced or disrupted lysosomal activity[49]. Yet, we found no presence of αSyn monomers within isolated lysosomes, possibly due to their rapid degradation by lysosomal proteases, making this possibility less likely. Macroautophagy of oligomers and/or larger aggregates from the cytoplasm, termed "aggrephagy", is a more likely mechanism, following from the well-recorded ability of autophagosomes to encapsulate cytoplasmic structures including aggregates[50], and target them to lysosomes.

Once released from the neurons, the fate of extracellular/interstitial αSyn aggregates also remains unclear, and can follow multiple possible scenarios: uptake into microglia[51], astrocytes[52], and/or oligodendrocytes[53], and/or drainage via glymphatics[54,55] or blood flow[6,56]. However, for the extracellular αSyn aggregates to cause the stereotypic "spread" of pathology via a prion-like etiology, other neurons will have to take up the released aggregates, and this process remains even less understood than the glial uptake or clearance scenarios.

The results presented here provide evidence of the accumulation of pathogenic αSyn species in neuronal lysosomes, and pinpoint to lysosomal exocytosis as a pathway of release for these pathogenic αSyn species into the extracellular milieu. This pathway, at first sight, appears to have relevance to the release of undegraded aggregates of other proteins as well that transmit pathology from cell-to-cell (reviewed in ref. [38]), and follow-up studies will be needed to test this speculation.

## Methods

### Animal research ethics statement
Animal husbandry and the experimental protocols used in this study were approved by the Institutional Animal Care and Use Committee (IACUC) at Weill Cornell Medicine (Animal protocol number: 2014-0015). The experiments also comply with the ARRIVE guidelines (https://www.nc3rs.org.uk/arrive-guidelines).

### CSPα knockout mice rescued by transgenic αSyn^A53T expression
Transgenic mice that express αSyn^A53T under the control of Thy-1 promoter[28] were crossed to CSPα knockout mouse line[29] by breeding Tg-αSyn^A53T to CSPα^+/- mice. The early neurodegeneration and death of CSPα^-/- mice was rescued by overexpressing αSyn^A53T [28]. Both lines were generated in C57BL/6 background and the cross has been inbred for >20 generations before experiments. Rescued Tg-αSyn^A53T/CSPα^-/- mice were bred with CSPα^+/- mice to obtain littermate CSPα^+/- and CSPα^-/- mice that either express or lack αSyn^A53T transgene. CSPα^+/- mice have no CSPα loss of function phenotype. Both, Tg-αSyn^A53T and CSPα^-/- mouse lines are available from Jackson Laboratory: B6.Cg-Tg(THY1-SNCA*A53T)M53Sud/J (stock # 008135) and B6.129S6-Dnajc5tm1Sud/J (stock # 006392), respectively.

### Generation of Tg-^HALAMP1^Myc mice
The transgenic mice were derived from a lentiviral vector. The lentiviral vector included truncated human synapsin-1 promoter followed by ORF including LAMP1 signal sequence−2xHA epitope tag−rat LAMP1−6His−TEV cleavage site−2xMyc epitope tag. This lentivirus vector was digested at 37 °C with PacI and SphI to create a 3.45 kb linear fragment for pronuclear microinjection. The cleavage product included truncated 5′ LTR, the synapsin promoter and ^HALAMP1^Myc ORF (as described above), WPRE, and part of the bGH poly(A) signal. The enzymes were heat inactivated at 65 °C for 20 min and the 3.45 kb band was purified/extracted using the Qiaquick gel extraction kit. The pronuclear injection was performed at Cornell University Stem Cell & Transgenic Core Facility into B6(Cg)-Tyrc-2J/JxFVB embryos. 115 embryos were injected, and 76 embryos proceeded to the 2-cell stage. Of the 18 clones born, 2 positive clones (male founders 1 and 7) contained PCR-detectable full-length ^HALAMP1^Myc, followed by confirmation by immunoblotting for protein expression. The founder transgenic clones were bred to wild-type female C57BL/6 mice and backcrossed >10 times. The genotyping primers amplify sequences either near the N-terminal end (F: 5′-CGCGACCATCTGCGCTG-3′ and R: 5′-GCTGTGCCGTTGTTGTC-3′; product = 297 bp), or near the C-terminal end (F: 5′-GCACATCTTTGTCAGCAAGGCG-3′ and R: 5′-GCAATAGCATGATACAAAGGC-3: product = 474 bp) of the insert.

### Crossing Tg-αSyn^A53T mice with Tg-^HALAMP1^Myc mice
Neuron-specific ^HALAMP1^Myc transgenic mice were crossed to Tg-αSyn^A53T mice, and bred either as single transgenics (Tg-^HALAMP1^Myc to Tg-αSyn^A53T) or bred as Tg-αSyn^A53T/Tg-^HALAMP1^Myc double-transgenic to wild-type. Either breeding-scheme produced Mendelian 25% Tg-αSyn^A53T/Tg-^HALAMP1^Myc and 25% wild-type progeny, and 25% progeny carrying each transgene alone; indicating single copy or insertion site for each transgene, carried on distinct chromosomes. Mice derived from a single pair cross (both generated in C57BL/6 background) were inbred further for 7 generations before initial experiments.

### αSynuclein knockout (αSyn^-/-) mice
Previously generated and characterized αSynuclein knockout (αSyn^-/-) mice generated and maintained on C57BL/6J background[57], and inbred for >20 generations, were used for primary neuron cultures.

### Lysosome (dextranosome) isolation from mouse brains via density gradient centrifugation
Enrichment of dextran-loaded lysosomes (dextranosomes), enhanced by mitochondrial swelling, was based on previously published techniques[58,59]. Mice were anesthetized with isoflurane and intracranially injected with 10 μl dextran-70 (250 mg/ml) into each cortex. 36 h later, mice were terminally anesthetized and perfused with homogenization buffer (HM = 10 mM HEPES buffer pH 7.0 with 0.25 M sucrose and 1 mM EDTA). The brain was dissected out, washed with HM and homogenized in 1.5 ml HM. Homogenate was centrifuged at $340 \times g_{av}$ for 5 min, any floating fat layer was aspirated, pellet discarded, and the supernatant was re-centrifuged at $340 \times g_{av}$ for 5 min. The resultant post-nuclear supernatant was incubated with 1 mM CaCl$_2$ for 5 min at 37 °C to swell the mitochondria, in order to reduce mitochondrial contamination in the dextranosomal fractions. This solution was centrifuged at $10,000 \times g_{av}$ for 30 min to precipitate heavy organelles. The pellet containing swollen mitochondria, peroxisomes and dextranosomes was resuspended in 1 ml HM and layered over 9 ml of 27% v/v Percoll in 0.25 M sucrose, followed by centrifugation at $35,000 \times g_{av}$ for 90 min, to generate the Percoll gradient. 1 ml fractions were collected from the top, with dextranosomes expected near the bottom of the gradient. Fractions were centrifuged at $100,000 \times g_{av}$ for 1 h to pellet the Percoll particles. The supernatant above the Percoll pellet was concentrated using Amicon (10 kDa cutoff), and assayed.

### Assays for cathepsin-D, citrate synthase, and catalase activity

Activities of enzymes contained within the lysosomes (cathepsin-D), mitochondria (citrate synthase), or peroxisomes (catalase) were measured from the Percoll gradient fractions dissolved in 0.1% Triton X-100 (final). Cathepsin-D activity was assayed using ab65302 kit (Abcam) according to the manufacturer's protocol. Each fraction was incubated with the cathepsin-D substrate GKPILFFRLK(Dnp)-D-R-NH$_2$ labeled with MCA at 37 °C for 1 h in the dark. Fluorescence was measured at Ex: 328 nm, Em: 460 nm, in a solid white 96-well plate (Costar). Citrate synthase activity was assayed using ab239712 kit (Abcam) according to the manufacturer's protocol, and absorbance at 412 nm was measured in clear bottom 96-well plate (Costar). Catalase activity was assayed using ab83464 kit (Abcam) according to the manufacturer's protocol, and absorbance at 570 nm was measured in clear bottom 96-well plate (Costar). Synergy H1 Hybrid Reader was used for all three enzyme assays.

### Lactate dehydrogenase cell death assay

Culture medium was changed 48 h prior to medium collection. Conditioned medium was subjected to lactate dehydrogenase (LDH) activity measurement using the ab102526 colorimetric assay kit (Abcam). LDH activity was also measured in parallel in plain medium (background) and in cells lysed in 1% Triton X-100. Same proportion of both the conditioned medium and cells were subjected to the LDH assay (1/20$^{th}$ of a well). After developing the colorimetric indicator for 30 min at room temperature in both, NADH standards and experimental samples, absorbance at 450 nm was measured using a Synergy H1 Hybrid Reader. LDH activity in samples was calculated based on the standard regression curve, and the LDH activity in conditioned media was plotted as percent of that in the cell lysate.

### Neuromuscular behavior tests and survival study

*Pole test*. The pole test was performed based on the method established by Ogawa et al.[60]. Animals were positioned with their head upward near the top of a wooden dowel (1 cm in diameter and 50 cm high). The time taken until they turned completely downward (defined as a "T-turn", indicative of bradykinesia) and time taken after the T-turn to descend to the bottom (indicative of motor coordination) were recorded. All animals were trained three times before performing the test. The maximum time allowed for each measurement was 20 s. If the animal could not turn, or fell during descent, the experiment was repeated; and if the animal again could not turn or fell during descent, the time for the activity was recorded as the maximum 20 s. *Grid hang test*. As in[61], animals were placed on top of a wire mesh grid (1.27 cm × 1.27 cm). The grid was then shaken lightly to cause the mouse to grip the wires with all four limbs, and then turned upside down. The mesh was held ~20 cm above the home cage litter, high enough to prevent the mouse from easily climbing down, but not to cause harm in the event of a fall. A stopwatch was used to record the time taken by the animal to fall off the grid. Three trials per mouse were performed with a 1 min inter-trial interval. The highest time of the 3 trials was assigned as the time for each animal to fall. After a maximum hang time of 120 s, mice were removed from the grid. *Hind limb clasping*. Mice were lifted by tail for 20 s. If an animal retracted its hind limbs toward the abdomen and held them there, it was scored positive for hind-limb clasping. *Survival study*. End point for survival study was reaching neuromuscular debility (e.g., paresis) which could make it difficult for the animal to get to food or water. At that time, the animal was euthanized according to the approved protocols.

### Immunoisolation of late-endosomes/lysosomes (LAMP1-positive) and lysosomes (TMEM192-positive) from mouse brains and primary neurons

Immunoprecipitation was used to isolate lysosomes from Tg-$^{HA}$LAMP1$^{Myc}$ and Tg-αSyn$^{A53T}$/Tg-$^{HA}$LAMP1$^{Myc}$ mouse brains. Mouse brains were lysed in 5 ml PBS without detergent and supplemented with EDTA-free protease-inhibitor cocktail (Thermo Fisher). The post-nuclear supernatant (500 μl; corresponding to 1/10 of a brain) was incubated with 50 μl anti-myc magnetic beads 50% slurry (with covalently cross-linked clone 9E10 mAb; Pierce) for 2 h at 4 °C, followed by 3 washes in 1 ml PBS by 1 min of tube-rolling at 4 °C (no vortex). The precipitated material was eluted in 25 μl of non-reducing 2× Laemmli sample buffer at room temperature, followed by immunoblotting.

A lentiviral construct encoding lysosomal protein TMEM192 tagged with 3xHA on the cytosolic C-terminus (PLJC5-Tmem192-3xHA[36,37] was expressed in primary neuron cultures by infecting them on DIV7. Neurons (10 cm dish) were harvested at 4 °C by freeze-thaw and pipetting in 1 ml PBS supplemented with EDTA-free protease-inhibitor cocktail (Thermo Fisher). This suspension was further homogenized using a microcentrifuge pestle for 1 min, followed by spinning down of heavy debris, such as unbroken cells and nuclei, at 500 g for 5 min. The post-nuclear supernatant was incubated with 10 μl anti-HA antibody for 2 h, then with 20 μl of 50% slurry of protein-G magnetic beads for 1 h at 4 °C, followed by 3 washes in 500 μl PBS. The precipitated material was eluted in 25 μl of non-reducing 2× Laemmli sample buffer at room temperature, followed by immunoblotting.

### Mouse CSF collection

6-month-old Tg-αSyn$^{A53T}$ mice were terminally anesthetized with isoflurane. CSF was collected by exposing cisterna magna[62], and using syringe aspiration (Hamilton 10 μl syringe). Samples with visible signs of blood were discarded. Following CSF collection, the mice were secondarily euthanized by cervical dislocation and brain collected for unrelated experiments. Due to low and/or variable volume of CSF collected in different attempts, 10–12 mice were tapped per experiment and the CSF was pooled to reach 80–100 μl total; followed by limited proteolysis performed on 9 μl of CSF per protease/detergent condition.

### Immunoprecipitation

For co-immunoprecipitation of SNARE proteins, brain homogenate from a single cortex was lysed in 5 ml of PBS containing 0.1% Triton X-100 and supplemented with EDTA-free protease-inhibitor cocktail (Thermo Fisher). 100 μl of post-nuclear supernatant was incubated for 2 h at 4 °C with 10 μl of the antibody, and then for another 1 h following the addition of protein-G Sepharose (GE Healthcare; 20 μl of a 50% slurry); or a multiple of this PNS: antibody: Prot.G beads ratio. Protein bound to antibody beads was washed five times with 500 μl lysis buffer at 4 °C using a vortex shaker (due to the low off-rate of SNARE interactions the washes can be harsher than typical co-IP experiments), and eluted in 20 μl of 2× Laemmli sample buffer. Eluent was boiled for 20 min to disassemble SNARE complexes, followed by immunoblotting.

### Long-term primary neuron culture from Tg$^{x2}$-αSyn$^{A53T}$ mice

Primary cultures were based on modification of our neuron culture protocol[35] with published procedures for long-term cultures[63]. Neonatal (P0) cerebral cortices were isolated and dissected in cold HBSS. Cortices were dissociated by trypsinization (0.05% trypsin-EDTA for 10 min at 37 °C), triturated with a siliconized pipette and plated onto poly-L-lysine-coated 24-well plastic dishes at high density (800–1000 cells/mm$^2$; each experiment had the same cell density across plates), in DMEM (Invitrogen) supplemented with 5% horse serum (Invitrogen) and 1% penicillin–streptomycin (Life Technologies). Humidity was enhanced by adding autoclaved water in spaces between the wells, and the plates were not disturbed for at least 5 days. On day 5–7, medium was changed to serum-free neurobasal medium (Life Technologies) supplemented with 2% B27 (Life Technologies) or N21-MAX (R&D Systems), and neurons were infected at the same time

with lentiviruses. 50% of the medium was changed every 7 days. Neuronal cultures were maintained for up to 7 weeks (DIV49).

Tg$^{x2}$-αSyn$^{A53T}$ mouse breeders were derived from Tg-αSyn$^{A53T}$ mice crossed to each other. However, this line was difficult to breed due to short survival and unreliable breeding output. Alternatively, we cultured cortices from all of the pups from either Tg-αSyn$^{A53T}$ parents, or from Tg$^{x2}$-αSyn$^{A53T}$ male bred with Tg-αSyn$^{A53T}$ females (harem breeding). The pups were tested for homozygosity by cerebellar immunoblots against αSyn. Only Tg$^{x2}$-αSyn$^{A53T}$ cultures were maintained through to the DIV7 steps of medium replacement and lentiviral infection.

Conditioned medium from neurons was collected either from 24 well (0.5 ml per well) or 6 well (2 ml per well) dishes, with maintenance of humidity by autoclaved water in spaces between the wells. Identical volume of medium was used throughout the repetitions of each experiment. Medium was spun through 5 µm mini strainer (Pluriselect) to exclude any debris, followed by concentration on 10 kDa Amicon (Millipore).

### Limited proteolysis of neuron culture media and CSF
Fresh pooled CSF or 10× concentrated media using 10 kDa cutoff Amicon (Millipore) at room temperature was aliquoted into 2 equal portions, and equal volume of either PBS or PBS containing Triton X-100 (0.1% final) was added to each tube and incubated for 10 min. Samples were then mixed with equal volume of Proteinase K (0, 1, 10, or 100 µg/ml final) from 100× stocks in sterile water, and incubated for 1 h at 15 °C (to minimize membrane permeability). Proteolysis was immediately stopped by adding 5× Laemmli sample buffer with PMSF (1 mM final), followed by immunoblotting.

### Limited proteolysis of immunoisolated late-endosomes/lysosomes
Goat anti-myc antibody (Abcam ab9132) was used for immunoisolation as indicated above, and proteolysis was performed on beads with addition of equal volume of Proteinase K at increasing concentrations from 100× stocks in PBS, and incubated for 1 h at 15 °C (to minimize membrane permeability). Proteolysis was immediately stopped by adding 5× Laemmli sample buffer with PMSF (1 mM final), followed by immunoblotting.

### Lentivirus production and transduction
HEK293T cells were co-transfected with the lentiviral vector plasmid and the HIV-1 lentiviral packaging constructs pRSVREV, pMDLg/pRRE[64], and the vesicular stomatitis virus-G expression plasmid pHCMVG[65]. The virus-containing culture supernatant was collected 48 h post-transfection and concentrated by centrifugation at 50,000 × $g_{av}$ for 90 min. The viral pellet was resuspended in neuronal medium (at 1/10 of the pre-centrifugation volume). All lentiviruses used in a single experiment were prepared together. Neurons were infected on DIV 5–7, at multiplicity of infection (MOI) of appx. 10, and harvested for experiments at times described in figure captions.

### Immunofluorescence and proximity ligation assay in neurons
For immunofluorescent labeling of cortical neuron cultures, cells on coverslips were washed with PBS + 1 mM MgCl$_2$ and fixed in 4% paraformaldehyde for 30 min at room temperature. Cells were permeabilized for 5 min in PBS + 0.1% Triton X-100, and blocked for 20 min in PBS + 5% BSA (blocking buffer). Coverslips were then incubated in primary antibodies in blocking buffer overnight at 4 °C. Following five washes, coverslips were incubated with secondary antibodies labeled with Alexa 488 and/or Alexa 546 (Life Technologies) in blocking buffer containing DAPI for 1–2 h, followed by five PBS washes and mounting on slides in Fluoromount G (Thermo Fisher). Images were acquired on a Nikon H550L microscope.

Tuj-1 positive neuronal soma and DAPI-stained nuclei were counted manually on coded and randomized slides. Each independent experimental "n" is represented by an average of four randomly selected regions on the coverslip.

For PLA experiments (DUO92105; Sigma), after washing primary antibodies, cells were then incubated with the minus and plus probes for 1 h at 37 °C and washed two times. Probes were then ligated for 30 min at 37 °C, washed two times in Buffer A, and amplified using the manufacturer's polymerase for 100 min at 37 °C in the dark. Coverslips were mounted on slides in Fluoromount G (Thermo Fisher), followed by image acquisition on a Nikon H550L microscope. PLA puncta were quantified with the image processing and analysis software ImageJ (National Institutes of Health). PLA puncta and DAPI puncta were counted in each image, and the PLA puncta were normalized to total DAPI puncta in the same image. Three images per "n" were averaged, where each $n$ = independent mouse pup, culture, and PLA experiment. Image acquisition and thresholding parameters were kept constant across each experiment.

For immunohistochemical studies, anesthetized mice were perfused with 4% paraformaldehyde in PBS, followed by removal of the brain and overnight fixation in 4% paraformaldehyde in PBS at 4 °C. Fixed brains were cryopreserved in 30% sucrose in PBS for 2 d and frozen in Tissue-Tek Optimal Cutting Temperature embedding medium (Sakura Finetek). Coronal brain sections (20 µm) were cut at −20 °C (Thermo Scientific Microm HM550 cryostat), picked up on slides, and heat adhered at 40 °C for 15 min. For immunostaining, slides were washed three times with PBS, incubated in 0.1% Triton X-100 in PBS for 1 h, washed two times with PBS, and blocked with 5% BSA for 30 min. Primary antibodies were incubated overnight at 4 °C. Slides were washed three times in PBS and blocked with 5% BSA for 20 min, followed by incubation with DAPI and secondary antibodies labeled with Alexa 488 and Alexa 555 (Life Technologies) for 2 h at room temperature. Following six washes in PBS, slides were mounted with Fluoromount G mounting medium (SouthernBiotech). Images were acquired on a Keyence fluorescence microscope (BZ-X710) at ×40 magnification and Z-stacked.

### Electron microscopy on mouse brains
Electron microscopy was used to identify αSyn aggregates within lysosomes of cortical neurons from Tg-αSyn$^{A53T}$ or WT mouse brains. Mice were deeply anesthetized (120 mg/kg pentobarbital) and perfused through the heart with a series of solutions (5 ml heparin (1000 U/ml), 30 ml of 3.75% acrolein/2% paraformaldehyde, followed by 20 ml of 2% paraformaldehyde) using a peristaltic pump (3 ml/min). The brain was coronally sectioned (40 µm) on a vibratome in 0.1 M phosphate buffer (PB) and sections (Bregma −3.40 mm) were processed for immunocytochemistry using a primary antibody cocktail against aggregated αSyn (Mouse 5G4 anti-aggregated αSyn, Millipore and Rabbit anti-αSyn filamentous, Abcam). For immunolabeling, all the tissue was co-processed and placed in 1% sodium borohydride in 0.1 M PB to remove excess aldehydes, rinsed, blocked with 0.5% BSA, and incubated for 48 h at room temperature in the primary antibody cocktail solution. Afterwards, the sections were rinsed and incubated for 1 h in colloidal gold IgG (1 nm; EMS), fixed in 2% glutaraldehyde, and enhanced with a silver solution (IntenS-EM kit: EMS) for 7 min. The tissue was postfixed with 2% osmium tetroxide for 1 h, dehydrated in a series of ethanols (30%, 50%, 70% 95%, 100%) and propylene oxide, and then embedded in plastic (Embed 812 Epon substitute; EMS). The embedded tissue was cut (70 nm) on an ultratome (Leica) using a diamond knife, collected onto mesh copper grids (EMS), and imaged using a transmission electron microscope (Hitachi HT7800). A total tissue area of 20,420 µm² (120 micrographs at ×8000–20,000 magnification) of cortical tissue was examined from 3 mice per group (WT or Tg-αSyn$^{A53T}$ mice). The αSyn aggregate density within lysosomes was calculated by counting the

number of individual immunogold particles within a lysosome and dividing it by the area of the lysosome. Quantification of neuronal lysosomes was ensured by solely including somata with visible synaptic contacts. The primary antibody-mixture used had minimal background level of staining in WT mouse brains, where no aggregates have been observed using multiple techniques, either in this or other studies. Unpaired 2-tailed Student's $t$ test and non-parametric Mann–Whitney test were used to analyze differences in αSyn aggregate gold density.

### SNARE shRNA knockdown and dominant-negative strategies

Once VAMP7 and SNAP23 were identified as cognate lysosomal SNAREs, in agreement with prior studies[40], we modulated their function via shRNA knockdown and dominant-negative strategies: Based on pre-characterized target sequences and hairpins described on the Broad Institute Genetic Perturbation Platform, mouse VAMP7 and SNAP23 shRNAs were cloned into L309 lentiviral vector to be expressed under human H1 promoter. For VAMP7, the target sequence was 5′-TTACGGTTCAAGAGCACAAAC-3′, and for SNAP23, the target sequence was 5′-CAACCGAGCCGGATTACAAAT-3′. Rescue expression of human SNAP23 from FUW vector[66] was not affected by the anti-mouse shRNA target sequence.

To express a dominant-negative VAMP7 fragment, rat VAMP7/TI-VAMP truncated at the C-terminal end (Ala$^2$-Asn$^{120}$ fragment)[67] was expressed as an eGFP chimera (GFP-VAMP7$^{2-120}$) via FUGW lentiviral vector[66].

To generate a dominant-negative version of SNAP23, the last eight residues of human SNAP23 were deleted (Asp$^2$-Ala$^{203}$ segment) mimicking the dominant-negative SNAP25$^{1-197}$ segment generated by Botulinum toxin-A cleavage[68] and expressed as an eGFP chimera (GFP-SNAP23$^{2-203}$) via FUGW lentiviral vector[66].

### Pharmacological manipulation of neuronal activity and cytosolic Ca$^{2+}$

Primary neurons at DIV48 were incubated for 24 h in fresh medium containing vehicle (0.1% DMSO), tetrodotoxin (TTX; 1 µM), amino-5-phosphonopentanoic acid (APV; 100 µM) + 6-cyano-7-nitroquinoxaline-2,3-dione (CNQX; 10 µM), KCl (10 mM), bicuculline methiodide (100 µM), dantrolene (Dan; 1 µM) + 2-Aminoethoxydiphenylborate (2-APB; 50 µM), or YM-58483 (10 µM). The media were collected and cells were lysed on DIV49, and analyzed as shown.

### Proximity-labeling of lysosomal contents

To target Apex-2 to the lysosomal lumen, Apex-2 was fused at the N-terminus to the transmembrane domain of rat LAMP1. The $^{Apex-2}$LAMP1 construct consists of: LAMP1 signal sequence—Flag tag—Apex-2—rat LAMP1 truncation including transmembrane domain plus C-terminal cytosolic tail with lysosome-targeting motif (LAMP1$^{370-407}$ including the targeting motif $^{403}$GYQTI$^{407}$)—6His—TEV—Myc tag, generating a 36 kDa protein.

$^{Apex-2}$LAMP1 was lentivirally expressed via a truncated human synapsin-1 promoter in Tg$^{x2}$-αSyn$^{A53T}$ primary neurons by infection on DIV7. On DIV49, cells were incubated at 37 °C for 1 h in medium containing 500 µM biotin tyramide, plus a lysosomal pH-raising cocktail (10 nM NH$_4$Cl and 100 nM Bafilomycin A1) to enhance the reaction. H$_2$O$_2$ was added at a final concentration of 1 mM for exactly 1 min at room temperature. Reaction was quenched for 30 sec with PBS + 1 mM MgCl$_2$ containing antioxidants (5 mM Trolox, 10 mM ascorbic acid). Cells were washed with PBS + 1 mM MgCl$_2$, and lysed in PBS + 0.1% Triton X-100, and biotinylated proteins were precipitated on streptavidin-magnetic beads (Dynabeads; Thermo), with 3 washes before elution in 2× Laemmli sample buffer.

To chase biotin-labeled proteins, Tg$^{x2}$-αSyn$^{A53T}$ primary neurons lentivirally expressing $^{Apex-2}$LAMP1, were subjected to biotin labeling on DIV 47, similar to above, except using 0.5 mM H$_2$O$_2$ to reduce toxicity.

The reaction was quenched with 37 °C medium containing antioxidants (5 mM Trolox, 10 mM sodium ascorbate, and 5 mM glutathione). Ten minutes later, the medium was replaced with medium containing 2.5 mM glutathione. 48 h later, the culture medium was collected, and biotinylated proteins were precipitated on streptavidin-magnetic beads (Dynabeads; Thermo), with 3 washes before elution in 2× Laemmli sample buffer.

### Purification and in vitro aggregation of recombinant myc-αSyn

Recombinant myc-αSyn was expressed and purified essentially as in[69]. Full-length human αSyn cDNA containing an N-terminal myc epitope-tag was inserted into a modified pGEX-KG vector (GE Healthcare), after a TEV protease recognition site to allow cleavage from GST (myc-αSyn contains an extra N-terminal glycine after cleavage with TEV protease). BL21 strain of *E. coli* bacteria transformed with this plasmid were grown to optical density 0.6 (at 600 nm), and protein expression was induced with 0.05 mM isopropyl β-ᴅ-thiogalactoside (IPTG) for 6 h at room temperature. Bacteria were pelleted by centrifugation for 20 min at 4000 rpm, and then resuspended in solubilization/lysis buffer: PBS containing 0.5 mg/ml lysozyme, 1 mM PMSF, DNase, and an EDTA-free protease-inhibitor mixture (Roche). Bacteria were further broken by sonication, and insoluble material was removed by centrifugation for 30 min at 7000 × $g_{av}$ at 4 °C. Protein was affinity-bound to glutathione Sepharose beads (GE Healthcare) by incubation overnight at 4 °C, followed by TEV protease (Invitrogen) cleavage for 6 h at room temperature. The His-tagged TEV protease was removed by incubation with Ni-NTA (Qiagen) overnight at 4 °C. Protein concentration was assessed using the bicinchoninic acid (BCA) assay (Thermo Fisher Scientific).

For the aggregation studies, 10 µl of concentrated extracellular medium from neuronal cultures—from 100 µl medium collected after 2 days of conditioning (DIV 47–49), spun through 5 µm strainer (Puriselect), and concentrated 10× using 10 kDa cutoff Amicon (Millipore)—was added to 40 µl of recombinant myc-αSyn (to 4 µg/µl final concentration) in PBS with protease inhibitors. This mixture was incubated at 37 °C with shaking at 300 rpm, while sample aliquots were taken at indicated time periods and frozen at −80 °C. At the end of the final incubation time-point, all the samples were thawed and measured together.

5 µl of sample was mixed with 95 µl of 100 µM K114 (Santa Cruz Biotechnology) in 100 mM glycine-NaOH, pH 8.45, and K114 fluorescence was measured at Ex: 390 nm, Em: 535 nm, in a solid white 96-well plate (Costar) using a plate reader (Synergy H1 Hybrid Reader, BioTek). Thioflavin-T measurement was done similarly, by mixing 5 µl sample with 95 µl of 25 µM Thioflavin-T (Sigma) in PBS, pH 7.4, and fluorescence was measured at Ex: 450 nm, Em: 485 nm. After fluorometry, the same samples were dissolved in Laemmli sample buffer and used for SDS-PAGE or dot blots, followed by immunoblotting.

### Serial extraction of primary neurons

Cells grown in 24-well plates (~85 µg total protein per well by BCA assay; Pierce), were serially extracted in 40 µl of buffers with escalating solubilization strength: First, the cells were frozen (−20 °C for 10 min) and thawed (4 °C for 20 min) in PBS, pH 7.4. The supernatant was collected after 100,000 × $g_{av}$ for 20 min. The pellet was solubilized in 0.1% Triton X-100 in PBS for 20 min, followed by centrifugation as above and collection of supernatant. The pellet was next solubilized in 1% SDS in PBS for 20 min, followed by centrifugation as above and collection of supernatant. The pellet was dissolved in 8 M urea in 2x Laemmli sample buffer at 37 °C for 2 h. All solubilization buffers, except urea, included protease-inhibitor cocktail (Roche), and were performed at 4 °C. Total protein was extracted directly in 2× Laemmli sample buffer. 1/2 of the well was separated for immunoblotting and 1/4th was dot-blotted.

## SDS-PAGE and quantitative immunoblotting

For SDS-PAGE, 10-15% Laemmli gels (10.3%T and 3.3%C) were used to separate proteins on Bio-Rad apparati. 1 mM final DTT was used in SDS Laemmli buffer. Samples separated had: 25 µg protein for brain or culture lysates, late endosomes/lysosomes isolated in parallel from equal amounts of brains/cultures, or media concentrated in parallel from equal amount of culture. Proteins were separated and transferred onto nitrocellulose (pore-size = 0.45 µm; GE Healthcare) and blocked with 5% w/v fat-free dry milk in tris buffered saline, pH 7.5 supplemented with 0.1% Tween 20 (TBS-T). Immunoblotting was performed by incubating the blocked membranes with primary antibodies in blocking buffer for 8–16 h. Following 5 washes with TBS-Tween 20 (0.1%). Blots were incubated with secondary antibodies (goat anti-rabbit conjugated to IRDye 600RD or 800CW; LI-COR) at 1/5000 in blocking buffer for 1–3 h. Immunoblots were washed 5× with TBS-T and dried, then scanned on Odyssey CLx (LI-COR) and quantified using Image-Studio software (LI-COR).

For ATP5G immunoblots, following the transfer, nitrocellulose membranes were dried and fixed for 15 min at room temperature in 1% paraformaldehyde in PBS[70]. Membranes were then washed 3x with TBS-T and treated as above.

For dot blots, 10 µg protein lysate, late endosomes/lysosomes isolated in parallel from equal amounts of brains/cultures, or media concentrated in parallel from equal amount of culture, were dotted onto dry nitrocellulose membrane (if sample volume <5 µl) or on a PBS wetted nitrocellulose membrane under vacuum (if volume was >5 µl) and allowed to dry. The membrane was then immunoblotted as above.

## Antibody list

β-Actin: Sigma (A1978); A11 amyloid: Stressmarq (SPC-506D); ATP5G: Abcam (ab181243); ATP6V1B2: Cell Signaling (D2F9R); ATP6V1E2: St. John's Labs (STJ191615); Calreticulin: Thermo Fisher (OTI15F5) and Novus (NB600-103); Cathepsin-D: R&D systems (AF1029); Cathepsin-L: Novus (JM10-78); CD81: Novus (SN206-01); EEA1: Thermo Fisher (MA5-14794); FLAG: Sigma Cl. M2 (F3165); GAPDH: Cell Signaling Cl. 14C10 (2118); GFP: Takara Cl. JL8 (632381) and Invitrogen (A11122); GluN2B/NR2B: Cell Signaling Cl. D15B3 (4212); GOSR1: Abcam (ab53288); GOSR2: ProteinTech (12095-1-AP); HA epitope: Abcam Cl. 16B12 HA.11 (ab130275); Rat HA (3F10; Sigma); Histone H3: Cell Signaling Cl. 96C10 (3638); HSP60: Abcam (ab5479); LAMP1: DSHB Cl. H4A3 and Cl. 1D4B DSHB Rat mono; LAMP2: Cl. ABL-93 DSHB Rat mono; MAP2: Millipore (AB5622); Myc epitope: DSHB (Cl. 9E10), Abcam (ab9132), Millipore (5G4), and Sigma (C3956); Na/K-ATPase: DSHB Cl. A6F; NeuN: Millipore Cl. A60 (Mab377); Neuroserpin: Abcam (ab33077); PEX3: Thermo Fisher (PA5-37012); PEX13: Sigma (ABC143); Sec22b: SYSY (186 003); Sec22c: Novus (NBP1-30760); Sec22L2: MyBiosource.com (MBS8507611); SNAP23: Santa Cruz (sc-166244) and Sudhof lab P914; SNAP25: Biolegend Cl. SMI81 (836304) and Sudhof lab P913; SNAP29: SYSY (111303); SNAP47: SYSY (111403); synaptogyrin-1: SYSY (103 002); αSyn: BD Transduction (610787) and Sudhof Lab T2270; αSyn filamentous (αSyn^Fila): Abcam Cl. MJFR-14-6-4-2 (ab209538); αSynuclein phosphorylated at Ser129 (αSyn^pSer129): Abcam (ab51253); TGN38: BD Transduction (610899); TIM23: BD Transduction (611223); TSG101: Abcam (ab125011); α-Tubulin: DSHB Cl. 12G10; VAMP1: SYSY (104 002); VAMP2: SYSY (104 202); VAMP5: SYSY (176 003); VAMP7/TiVAMP: SYSY (232 011) and SYSY (232 003); VAMP8: SYSY (104 302); VTI1A: SYSY (165 002); VTI1B: SYSY (164 002); Ykt6: Abcam (ab241382).

## Statistical analyses

Each "n" consisted of reagents produced in parallel (e.g., purified proteins, lentivirus production, culturing neurons, brain or CSF harvested from mice euthanized together) and experiments performed in parallel (e.g., lentiviral infection, sample collection for immunoprecipitation and immunoblotting, mouse behavior recorded on same day etc.). Experiments were quantified using parametric as well as non-parametric statistical tests when warranted (see the tests described below and in figure captions). No samples or animals were excluded from the analysis, and the investigators were blinded toward the genotype, virus infected into wells, pharmacological treatment, or coverslips/slides visualized. Randomization and coding were done by different investigator(s) than the investigator who did experiment and quantification/analysis. All analyses are performed using GraphPad Prism 8 and described in the figure legends. Column analyses of data with more than 2 groups depicted in bar graphs (e.g., αSyn species in lysosomal fractions and αSyn species in extracellular media) were analyzed by 1-way ANOVA with Dunnett multiple-comparison correction, as well as using the non-parametric Kruskal–Wallis test with Dunn's multiple-comparison adjustment. For comparison of only two groups (e.g., such as for littermate ratio of WT and Tg-^HALAMP1^Myc cross) paired 2-tailed Student's t test was used. For time-course experiments (e.g., accumulation of αSyn over age and seeding of recombinant αSyn) the curves were analyzed using 2-way ANOVA in comparing the overall curve, plus the Bonferroni post-test with multiple hypothesis correction to compare data at each time-point independently. For proteolysis experiment, the curves were analyzed via 2-way ANOVA with Bonferroni multiple comparisons post-test for looking at differences at each concentration. For 2-way ANOVAs, the main effect of the difference between the groups (e.g., ∓TX-100, genotypes) are reported in each figure. Graphs depicting the correlation (e.g., between the levels of pathogenic αSyn species released versus the levels of cathepsin-L released in the medium) were analyzed via Pearson's correlation and shown with linear regression lines, as well as the 95% confidence interval using dotted lines. Kaplan-Meier curves (e.g., survival, onset of hind-limb clasping, and onset of grid-hang impairments) were compared via Log-rank (Mantel–Cox) test. For the behavior experiments, each mouse was matched by age because the same mouse is repeatedly measured. For vertical pole studies, due to decreasing sample size with each mouse death, two separate curves were graphed to represent each of the pole-turn and the pole-descent data. Mice that died were scored as 20 s (maximum measurement) for rest of the trial—not as an actual measurement, but as a place holder. Data with 20 s place holder were analyzed by RM 2-way ANOVA, and data without the 20 s place holder were analyzed via mixed effects analysis due to missing data points when a mouse died. See Supplementary Table 1 for figure-by-figure statistical details.

## Reporting summary

Further information on research design is available in the Nature Research Reporting Summary linked to this article.

## Data availability

All data generated or analyzed during this study are included in this published article plus its Supplementary Information. Source data are provided with this paper.

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

## Acknowledgements

We thank Dr. Tom Südhof for kindly sharing the CSPα knockout and Tg-αSyn[A53T] mouse lines, as well as antibodies against αSyn, SNAP23, and SNAP25. This work was supported by NIH F31 studentship (NS098623, to N.N.N.), grants from Alzheimer's Association (NIRG-15-363678 to M.S.), American Federation for Aging Research (New Investigator in Alzheimer's Research Grant, to M.S.), and National Institutes of Health (1R01NS102181, 1R01NS113960 and 1R01NS121077 to J.B.; 1R01NS095988 and 1R01AG052505 to M.S.; and RF1NS126342 to J.B. and M.S.).

## Author contributions

Y.X.X., N.N.N., J.F., P.K., Y.N., D.L., J.B., and M.S. designed, performed and analyzed all experiments. Y.X.X., N.N.N., J.B., and M.S. wrote the paper. M.S. conceived the project and directed the research.

## Competing interests

The authors declare no competing interests.

## Additional information

**Supplementary information** The online version contains

supplementary material available at https://doi.org/10.1038/s41467-022-32625-1.

