## [Peer Review File · Nature Communications]

Lysosomal Exocytosis Releases Pathogenic α -Synuclein Species from NeuronsREVIEWER COMMENTS

Reviewer #1 (Remarks to the Author):

In the present study, Xie et al. suggest that lysosomal exocytosis is implicated in the cellular release of pathogenic alpha-synuclein (aSyn) species from neurons. Using transgenic mice expressing human A53T aSyn under the control of Thy-1 promoter, the authors show high molecular weight signal in aSyn immunoblots from endolysosome-enriched fractions of mouse brain or cortical primary neurons. Conditioned medium from A53T primary neurons expressing Apex-2Lamp1, revealed biotinylated high molecular weight aSyn species, likely independent of exosome/extracellular vesicle release. Release of endolysosomal proteins was SNARE-dependent, since expression of a dominant-negative form of VAMP7 (VAMP7DN) and knockdown of SNAP-23 (SNAP-23KD), showed a significant reduction of aSyn high molecular weight signal. Finally, aggregation of recombinant purified aSyn was faster during incubation with conditioned medium from A53T transgenic as compared to VAMP7DN/SNAP-23KD and wild type neurons. Based on this evidence, the authors conclude that potentially pathogenic forms of aSyn are associated with neuronal lysosomes and released via lysosomal exocytosis. The investigation of mechanisms of cellular release of pathogenic aSyn forms is a central focus in synucleinopathies, and understanding the contribution of lysosomal exocytosis has translational implications. Moreover, the established methods and generated animal model are well-controlled, and highly relevant for studying lysosomal biology in neurodegeneration. However, key aspects of the study need to be supported by further experimental evidence.

Major Concerns

- A central claim of this study is that neuronal lysosomes accumulate oligomeric/aggregated forms of aSyn, while monomeric aSyn is not detectable. To this end, ultrastructural evidence supporting the association of aSyn with endolysosomes should be provided. Moreover, specificity of the high molecular weight signal in immunoblots for aSyn requires validation by additional controls/methods. To this end, the authors may consider including preparations of whole brain and dextranosome fractions from aSyn knockout mice, and the sequential extraction of mouse brain lysate in TBS/SDS/Urea fractions.
- Moreover, it is challenging to determine individual aSyn species from the dot blot data, due to limitations in distinguishing different aSyn species in cell or tissue homogenates (Kumar et al. 2020). Complementarily, the presence of amyloid accumulations can be shown by additional methods such as immunofluorescence. Please provide Information on which antibody was used to detect amyloid-like aSyn.
- Based on the data presented on Fig. 2a, it is suggested that up to 20% of aggregated aSyn in the transgenic mouse brain is associated with neuronal lysosomes. However, normalization for this data is challenging, since the representation of neurons/glia in total brain homogenate, and recovery of total neuronal lysosomes upon immunolysis can be variable. An alternative would be to normalize against HALamp1Myc levels in their samples. Along those lines, the expression and subcellular distribution of HALamp1Myc at 1 and 6 months of age should be examined. For direct quantitative comparison of aSyn levels, a quantitative assay such as ELISA could be used. Lamp1 levels across samples should also be included for Fig. 2b, with the caveat that Lamp1 also measures late endosomes.
- In Fig. 4c, it is unclear how many independent primary neuron cultures were tested. It would be important to examine independent preparations, and include a control of unconditioned medium incubated for 28 days with recombinant aSyn. Please specify the number of conditioned neurons and initial volume of harvested conditioned medium. It would also be important to address viability of these cultures at the time of harvest.
- To demonstrate inhibition of the dynamic process of lysosomal exocytosis due to

VAMP7DN/SNAP23KD in neurons, stimuli such as inducers of lysosomal Ca²⁺ release, known to drive lysosome fusion to the plasma membrane, can be used. Potential approaches for visualizing lysosomal exocytosis include fluorogenic lysosomal enzyme substrates, or monitoring of the externalization of the luminal domain of lysosome-associated membrane proteins.

- **The authors suggest that the mechanism of lysosomal exocytosis is primarily accountable for releasing potentially pathogenic aSyn species. To elucidate this important point further, it would be interesting to compare the aggregation propensity of recombinant aSyn, by omitting (5µm filtration) or including (without filtration) the exosome/extracellular vesicle fraction in the paradigm of Fig. 4C. It could be argued that several constitutive pathways may simultaneously contribute to aSyn release, including background toxicity, which is unavoidable in cell culture. Thus, concluding on a single essential and rate-determining pathway for release of toxic forms of aSyn is challenging.**

- **The authors discuss that since no monomeric aSyn was detectable in isolated endolysosomes, higher-order forms of aSyn may reach neuronal lysosomes via aggrephagy. However, this scenario would require the progressive accumulation of prominent Lewy body-like aSyn pathology in THY1-SNCA*A53T mouse neurons. If this is not the case, additional mechanisms can be included in the discussion.**

Minor Concerns

- **LAMP1 is not exclusively a marker for mature lysosomes. The protein may also be present in mature endosomes and synaptic vesicle precursors.**

- **Please report ANOVA in detail.**

- **The integration of Fig. S1 in the context of this study could be improved. The data is interesting yet the implication of CSPα for lysosomal exocytosis of aSyn aggregates is not further experimentally addressed in the study.**

- **The authors indicate that aSyn species released upon lysosomal exocytosis are pathogenic, without, however, directly addressing pathogenicity e.g. by including paradigms of uptake/seeding in target cells. Please revise phrasing accordingly.**

- **Fig. S4f requires quantification, and whether all samples were run on the same blot needs clarification. Also, please revise the figure legend "... aSyn aggregates are detected in Tgx2-αSynA53T brains much earlier than in Tgx2-αSynA53T brains".**

- **The authors discuss literature suggesting that the majority of extracellular aSyn is not in association with exosomes/extracellular vesicles. The Discussion would benefit from including key studies demonstrating the association of aSyn with exosomes/extracellular vesicles in the human cerebrospinal fluid, and that uptake of this fraction is capable of leading to aSyn-related pathology in recipient cells. Based on the current knowledge, it would be challenging to conclude on the relative contribution of each of the proposed pathways to aSyn release/secretion.**

Reviewer #2 (Remarks to the Author):

The manuscript addresses the mechanism of a-synuclein secretion using in vivo and in vitro models, concluding that lysosomal exocytosis and VAMP7/ SNAP23 proteins are required for this process. The work addresses an important question in the field and uses relevant models to test their hypothesis. Most of the work is technically well done: they present clean data that shows successful isolation of endolysosomal vesicles using multiple techniques, and the rescue data with VAMP7 DN / SNAP23 KD is impressive. The nature of the secreted a-synuclein was also addressed well by in vitro seeding experiments. However, the experimental design could be improved. For example, there are a few critical aspects of the work that are conspicuously missing. This includes 1) The use of non-Tg wild-type mice to match with A53T; 2) measurements of cell death or lysis as controls, to rule them out as a source of extracellular lysosomal secretion, 3) lack of consideration for LAMP1 localization in endosomes, which is a source of exosomes.

Furthermore, while the biochemistry is well done, the use of imaging techniques such as immuno-electron microscopy would go a long way in providing direct evidence for a-synuclein accumulation in lysosomes and secretion via lysosomal exocytosis.

The link to CSP^{-/-} and +/- was presented early in the manuscript, but then faded away without any apparent reason. It would be interesting to follow through with these studies, for example by incorporating them into the lysosomal exocytosis studies. Perhaps the phenotype would be exacerbated. Without this, the data seem a little disjointed.

Other specific comments are indicated below.

1) The results of figure 1 are potentially interesting, but it is a little unusual to find only aggregate forms of a-synuclein within isolated lysosomes. Monomers are known to be internalized by CMA or delivered via macroautophagy, and at least some aggregates will break down into their monomeric components during SDS-PAGE, so one would expect some detectable amounts of monomers in these fractions, or at least proteolytic fragments. Perhaps a longer exposure would bring these out, or a stronger a-synuclein antibody that also detects mouse a-synuclein, and with the total lysate loaded on the same gel. If monomers cannot be detected, then other fractions should be analyzed to show where it is, perhaps cytosol fractions.

2) Conformation of aggregated a-synuclein with lysosomes should be done by immunohistochemistry or immuno-EM.

3) Analysis of mice expression endogenous wt a-synuclein should be included in figure 1, including lysosomal and cytosolic fractions.

4) I am not sure that I understand why or how dead mice could have been included in the behavioral test data of Figure S3f/g, S4d/e. Perhaps other behavioral tests could provide be a better readout. It would be important to know if overexpression of LAMP1 influenced the baseline A53T genotype.

5) There is no indication of how many times the new LAMP1 -A53T mice were backcrossed prior to doing experiments to assure that each of the 4 lines are comparable in terms of their genetic background.

6) In figure 2, the 1 month control is informative, but it will be essential to include a LAMP1 Tg expressing wild-type endogenous mouse a-synuclein in this experiment.

7) Neuronal health should be assessed in primary culture experiments of DIV 14 and 49, to determine if toxicity occurs during the long term culture. LAMP1 or another lysosomal marker in addition to CathL should be included.

8) Figure S6 is interesting, however it requires WT and non-aged tg controls. It would be critical to assess contamination by other material such as blood cells etc to make sure the signal is derived only from CSF.

9) Cell lysis or death needs to be excluded as a potential source of extracellular a-synuclein

10) LAMP1 does not exclusively label lysosomes but also late endosomes and a small portion of early endosomes (Saftig and Klumperman, 2009). The endosomal system feeds vesicles into multivesicular bodies (MVBs), which are the main source of exosomes (Trajkovic et al Science 2008). Therefore it is possible that the source of a-synuclein is from exosomes. It also somewhat surprising that tsg101 isn't biotinylated by a LAMP1 conjugate, given its clear association with the endosomal system. Perhaps tsg101 is not exposed to the lumen. More direct experiments would be required prior to ruling out an exosomal source of a-synuclein.

11) Throughout the manuscript, the MW of Cath L should be included, as it is composed of heavy and light chains. Determination of Cath L enzyme activity or other hydrolases in the media would strengthen the conclusions that the media contents are derived from

lysosomes.

12) In Figure 3B, it is possible that VAMP7 DN or SNAP23 KD are improving cell health, resulting in less a-synuclein in the media. This possibility could be easily ruled out with a cell death assay.

13) It would be important to know if WT neurons secrete any a-synuclein, since the media was used as a control in figure 4A, B. Immunodepleting a-synuclein from the media would be an important control here. If amyloidogenic seeds are formed and secreted, then sample agitation should not be required to stimulate recombinant myc-a-synuclein aggregation.

Reviewer #3 (Remarks to the Author):

This study does a nice job of documenting the lysosomal localization of pathogenic forms of aSyn, which was proposed previously by several groups. The original part of the study is the evidence indicating that SNARE-mediated lysosomal exocytosis is involved in the release of aSyn aggregates. However, lysosomal exocytosis has been extensively demonstrated in other cell types to be absolutely dependent on calcium transients in the donor cells, and the authors do not even discuss this. Is calcium the trigger? As presented, the study is incomplete because it also does not clarify how the exocytosis of aSyn aggregates leads to cell-cell transfer.

Main points:

1) In the introduction, the authors refer to Abounit et al. EMBO J 2016 as describing the role of tunneling nanotubes on trafficking of aSyn aggregates. However, they omit something important in their statement: the Abounit paper showed that that >90% of aSyn fibrils co-localized with lysosomal markers, and proposed that lysosomes were the organelles transferred from cell-to-cell through nanotubes. There seem to be additional studies showing that aSyn accumulates in lysosomes (Sung 2001, Lee 2008, Liu 2009, Hasegawa 2011). Thus, this statement in the introduction is somewhat misleading: "we present our finding that pathogenic aSyn species accumulate within the neuronal lysosomes in mouse brains and in primary neurons". The authors may have demonstrated more precisely the lysosomal localization using fractionation and proximity labeling methods, with good distinction between monomers and pathogenic species - but they must make clear which of their results are confirmatory of previous studies, and which are absolutely novel (such as release through lysosomal exocytosis).

2) In Fig. 3a, which fraction of the total pathogenic aSyn species present in the cells was released into the supernatant? Is this fraction similar for Cath-L and aSyn?

3) The authors cite Rao et al JBC in support of the involvement of VAMP7 and SNAP23 in lysosomal exocytosis. However, they omit something important – that study involved calcium-triggered lysosomal exocytosis. What is the role of calcium in their system? This is critical because only very low levels of lysosomal content release is observed, in many cell types, in the absence of calcium influx. They make a distinction on page 4 between lysosomal exocytosis of cathepsin L and constitutive exocytosis of neuroserpin, but do not explain what they mean by this. If not constitutive, is lysosomal exocytosis and pathological aSyn release calcium dependent? Can exocytosis be increased by inducing calcium transients in the cells, and blocked by calcium chelators?

4) The in vitro seeding of aSyn aggregates by supernatants containing secreted pathogenic aSyn is intriguing, but does not explain at all how these aggregates enter other cells. Are they taken up by endocytosis or phagocytosis? If they are, this would explain how they arrive in lysosomes, without a need to invoke autophagy. Without clarifying this aspect, the study is incomplete and does not advance the field much beyond previous publications.

Minor point:

The Ejlerskov 2013a and 2013b references are exactly the same.

REVIEWER COMMENTS

Reviewer #1 (Remarks to the Author):

In the present study, Xie et al. suggest that lysosomal exocytosis is implicated in the cellular release of pathogenic alpha-synuclein (aSyn) species from neurons. Using transgenic mice expressing human A53T α Syn under the control of Thy-1 promoter, the authors show high molecular weight signal in aSyn immunoblots from endolysosome-enriched fractions of mouse brain or cortical primary neurons. Conditioned medium from A53T primary neurons expressing Apex-2Lamp1, revealed biotinylated high molecular weight aSyn species, likely independent of exosome/extracellular vesicle release. Release of endolysosomal proteins was SNARE-dependent, since expression of a dominant-negative form of VAMP7 (VAMP7DN) and knockdown of SNAP-23 (SNAP-23KD), showed a significant reduction of aSyn high molecular weight signal. Finally, aggregation of recombinant purified aSyn was faster during incubation with conditioned medium from A53T transgenic as compared to VAMP7DN/SNAP-23KD and wild type neurons.

Based on this evidence, the authors conclude that potentially pathogenic forms of aSyn are associated with neuronal lysosomes and released via lysosomal exocytosis. The investigation of mechanisms of cellular release of pathogenic aSyn forms is a central focus in synucleinopathies, and understanding the contribution of lysosomal exocytosis has translational implications. Moreover, the established methods and generated animal model are well-controlled, and highly relevant for studying lysosomal biology in neurodegeneration.

We thank this reviewer for finding our study well-controlled and highly relevant.

However, key aspects of the study need to be supported by further experimental evidence.

We have added several new experiments to bolster our findings, as outlined below.

Major Concerns

- A central claim of this study is that neuronal lysosomes accumulate oligomeric/aggregated forms of aSyn, while monomeric aSyn is not detectable. To this end, ultrastructural evidence supporting the association of aSyn with endolysosomes should be provided.

We had previously included biochemical evidence of ultrastructural co-location by Apex-2 proximity-labeling, which labels molecules within an ultrastructural distance of 10-20nm (Hung et al., 2016; Mayer and Bendayan, 1997) (**Fig. 2b**). Now we have added visual evidence, using an *in-situ* proximity ligation assay which also labels ultrastructurally with a maximum distance of 40nm between two molecules (Bagchi et al., 2015) (**new Fig. 2c**), to confirm the presence of α Syn aggregates in lysosomes.

Moreover, specificity of the high molecular weight signal in immuno-blots for aSyn requires validation by additional controls/methods. To this end, the authors may consider including preparations of whole brain and dextranosome fractions from aSyn knockout mice, and the sequential extraction of mouse brain lysate in TBS/SDS/Urea fractions.

We have now included α Syn KO and WT neuron blots using detergent extraction (**new Suppl. Fig S5c-d**). Our data show SDS-extractability of α Syn aggregates, which are disrupted into monomers in presence of urea. These new blots support the specificity of the detected HMW signal by multiple separate methods: (1) High molecular mass signals were measured using two different antibodies: α Syn and pSer129 α Syn. These signals directly and quantitatively correlate with each other throughout the study. (2) These two measurements of high molecular mass signals also quantitatively correlate with dot blots against α Syn filament and amyloid-fold throughout the study. (3) Importantly, all of these four measurements were absent from pre-aggregate transgenic neuron cultures (**Fig. 2b, and Suppl. Figs. S5a & S6**) and mouse brains (**Fig. 2a, and Suppl. Figs. S1 & S4f**), as well as from WT neuron cultures (**new Suppl. Fig. S5c-d**) and mouse brains (**Fig. S4f**).

- Moreover, it is challenging to determine individual aSyn species from the dot blot data, due to limitations in distinguishing different aSyn species in cell or tissue homogenates (Kumar et al. 2020). Complementarily, the presence of amyloid accumulations can be shown by additional methods such as immunofluorescence. Please provide Information on which antibody was used to detect amyloid-like aSyn.

We agree with the weaknesses of dot-blot, if shown alone. Therefore, we have also included high molecular mass α Syn aggregates as well as pSer129-modified α Syn immunoblots throughout the study along with the dot blots, and found both signals to highly correlate. Yet, dot blots are important, since they are used here to detect large α Syn aggregates, which do not enter the gel fully due to their size. Dot blotting is well-accepted for detecting such species (Fontaine et al., 2016; Fusco et al., 2017; Guo et al., 2020; Lassen et al., 2018; Lindersson et al., 2004; Nuber et al., 2013; Paleologou et al., 2009) and others), including amyloid-like folds of α Syn (detected by A11 antibody against the peptide backbone in amyloid conformations; Stressmarq (SPC-506D)) and filamentous α Syn (Abcam Cl. MJFR-14-6-4-2 (ab209538)) developed and validated by the Michael J. Fox Foundation. In agreement with Kumar et al. 2020 and the reviewer, we have avoided calling any antibody oligomer-specific throughout the manuscript, since anti-“oligomer” antibodies do often recognize larger α -syn fibrils. These and all other antibodies used are listed in the Methods section under “Antibody list”.

In addition, as suggested by this reviewer, we now visualize the α Syn filaments in neurons via immunofluorescence (**New Suppl Fig. S5b**), and their ultrastructural location in lysosomes via proximity ligation (**New Fig. 2c**).

- Based on the data presented on Fig. 2a, it is suggested that up to 20% of aggregated α Syn in the transgenic mouse brain is associated with neuronal lysosomes. However, normalization for this data is challenging, since the representation of neurons/glia in total brain homogenate, and recovery of total neuronal lysosomes upon immuno-isolation can be variable. An alternative would be to normalize against HALamp1Myc levels in their samples. Along those lines, the expression and subcellular distribution of HALamp1Myc at 1 and 6 months of age should be examined. For direct quantitative comparison of α Syn levels, a quantitative assay such as ELISA could be used. Lamp1 levels across samples should also be included for Fig. 2b, with the caveat that Lamp1 also measures late endosomes.

We had actually normalized the quantifications as suggested by the reviewer (to neuronally expressed $^{HA}Lamp1^{Myc}$ as detected by HA antibody, not to lysosomal markers that may come from all cells). We apologize for the lack of clarity, and have now clearly noted this in the figure legend. The other lysosomal markers shown are to confirm the immunoisolation of lysosomes.

Additionally, $^{HA}LAMP1^{Myc}$ expression levels in transgenic mouse brains are shown in **Suppl. Fig. S2a** by immunoblots, and comparison between 1 and 6 months of age is shown in the ‘total lysate’ lanes in **Fig. 2a**. Immunofluorescence in **Suppl. Fig. S2a** is shown to visualize sub-cellular distribution of $^{HA}LAMP1^{Myc}$, supporting lysosomal localization. As suggested by the reviewer, we have now included Lamp1 and Lamp2 in **Fig. 2b**.

Also, as suggested by the reviewer, we have now mentioned in the manuscript that Lamp1 is also in late-endosomes and lysosomes when we introduce the Lamp1 constructs used for labeling and isolation studies. Furthermore, lysosomal accumulation of α Syn aggregates is bolstered by multiple Lamp1-independent techniques: Isolation of TMEM192-3xHA positive lysosomes (**new Suppl. Fig. S6**; characterized and used in (Abu-Remaileh et al., 2017; Wyant et al., 2018)), visual ultrastructural evidence, using *in-situ* proximity ligation assay (**new Fig. 2c**), and the previously shown dextranosome-isolation studies (**Fig 1**).

- In Fig. 4c, it is unclear how many independent primary neuron cultures were tested. It would be important to examine independent preparations, and include a control of unconditioned medium incubated for 28 days with recombinant α Syn. Please specify the number of conditioned neurons and initial volume of harvested conditioned medium. It would also be important to address viability of these cultures at the time of harvest.

These experiments were done using 4 independent cultures. Now we have included the word “independent” in the figure legend to clarify this. We have also now added new experiments including unconditioned medium (**new Suppl. Fig. S10b**).

However, it is important to note that WT mouse neurons do not have α Syn aggregates (**Suppl. Fig. S4f and new Suppl. Fig. S5c-d**). Thus, the addition of media from WT neurons is used as a control to produce a basal rate of aggregate formation. The experiment is not centered on the absolute rate of aggregate formation, but on the enhancement of this basal rate by the addition of α Syn aggregates released into the media from Tg^{x2}- α SynA53T neurons, and the reduction in this effect when the Tg^{x2}- α SynA53T neurons express VAMP7^{DN}.

We have noted the initial volume of conditioned media harvested in “Methods” section. Regarding viability of the cultures, actin and monomeric α Syn were shown to indicate a lack of leakage of cytoplasmic content (**Fig. 4b**). In addition, we have now included cell viability/death assays (**new Suppl. Fig. S5e-f**). We also show that reduced secretion of α Syn aggregates due to SNAP-23 KD and Vamp7 DN are not caused by cytoprotection (**new Suppl. Fig. S9d**).

- To demonstrate inhibition of the dynamic process of lysosomal exocytosis due to VAMP7DN/SNAP23KD in neurons, stimuli such as inducers of lysosomal Ca^{2+} release, known to drive lysosome fusion to the plasma membrane, can be used. Potential approaches for visualizing lysosomal exocytosis include fluorogenic lysosomal enzyme substrates, or monitoring of the externalization of the luminal domain of lysosome-associated membrane proteins.

We have now included the effect of neuronal activity and cytosolic calcium on the lysosomal exocytosis of α Syn aggregates (**new Suppl. Fig. S8**), and find that cytosolic calcium and neuronal activity do enhance lysosomal exocytosis of α Syn aggregates in neurons. The physiological regulation of lysosomal exocytosis is a fascinating subject and should be explored further in future studies.

- The authors suggest that the mechanism of lysosomal exocytosis is primarily accountable for releasing potentially pathogenic aSyn species. To elucidate this important point further, it would be interesting to compare the aggregation propensity of recombinant aSyn, by omitting (5 μ m filtration) or including (without filtration) the exosome/extracellular vesicle fraction in the paradigm of Fig. 4C. It could be argued that several constitutive pathways may simultaneously contribute to aSyn release, including background toxicity, which is unavoidable in cell culture. Thus, concluding on a single essential and rate-determining pathway for release of toxic forms of aSyn is challenging.

Exosomes, and even extracellular vesicles, are not excluded here by 5 μ m filtration (Kornilov et al., 2018). In addition, any exosomal or vesicular contents should not be templating purified α Syn aggregation from inside the membrane envelope of the exosome/vesicle. However, we have now clarified in the Discussion that other pathways of α Syn aggregate release cannot be excluded, reflecting the reviewer’s point/suggestion.

- The authors discuss that since no monomeric aSyn was detectable in isolated endolysosomes, higher-order forms of aSyn may reach neuronal lysosomes via aggrephagy. However, this scenario would require the progressive accumulation of prominent Lewy body-like aSyn pathology in THY1-SNCA*A53T mouse neurons. If this is not the case, additional mechanisms can be included in the discussion

Thank you for pointing this out. Our language regarding this topic in the Discussion was unclear. The presence of prominently large Lewy body-like inclusions is not required for aggrephagy or for higher order forms of α Syn to accumulate inside lysosomes. We have now changed the wording to reflect this. Additional possible/proposed mechanisms such as aggregation of monomers within the acidic environment of lysosomes are also included.

Minor Concerns

- LAMP1 is not exclusively a marker for mature lysosomes. The protein may also be present in mature endosomes and synaptic vesicle precursors.

As answered above, we have now indicated that Lamp-1 is present in late-endo/lysosomes in the text. Additionally, we now show α Syn aggregates in lysosomes by multiple Lamp1-independent techniques: (1) Isolation of TMEM192-3xHA positive lysosomes (**new Suppl. Fig. S6**; characterized and used in (Abu-Remaileh et al., 2017; Wyant et al., 2018)), (2) visual ultrastructural evidence, using *in-situ* proximity ligation assay (**new Fig. 2c**), and (3) the previously shown dextranosome-isolation studies (**Fig 1**).

- Please report ANOVA in detail.

Thank you for pointing this out. Now additional details are included in the figure legends and “Statistical analyses” section in “Methods”. A detailed table of figure-by-figure ANOVA analyses is added under the “Statistical analyses” section.

- The integration of Fig. S1 in the context of this study could be improved. The data is interesting yet the implication of CSP α for lysosomal exocytosis of aSyn aggregates is not further experimentally addressed in the study.

This experiment was our initial indication of the involvement of lysosomes, showing a lysosome-storage defect produced by the CSP α null phenotype correlating with accumulation of α Syn aggregates. To address the specific question of whether α Syn aggregates accumulate in neuronal lysosomes, we had to deconvolute the model systems by removing the severe lysosomal storage phenotype produced by CSP α KO. We are continuing to study the CSP α knockout mouse to understand the cellular mechanism of the CSP α mutation-linked lysosomal storage disease (adult-onset neuronal ceroid lipofuscinosis; ANCL), independent from α Syn aggregate accumulation and exocytosis.

- The authors indicate that aSyn species released upon lysosomal exocytosis are pathogenic, without, however, directly addressing pathogenicity e.g. by including paradigms of uptake/seeding in target cells. Please revise phrasing accordingly.

The term “pathogenic species” for high molecular mass aggregates of α Syn that are phosphorylated at Ser129 and recognized by antibodies against α Syn filaments and against amyloid-folds is firmly based on a large amount of literature from studies in human patients and mouse models, as well as from cell biological studies (Anderson et al., 2006; Elfarrash et al., 2019; Fujiwara et al., 2002; Kim et al., 2019; Lau et al., 2020; Luk et al., 2012; Mao et al., 2016; Volpicelli-Daley et al., 2011). In support of this terminology, we have now added an experiment testing seeding of aggregates in neurons by the released α Syn species (**new Fig. 5d**).

- Fig. S4f requires quantification, and whether all samples were run on the same blot needs clarification. Also, please revise the figure legend “... α Syn aggregates are detected in Tg α 2- α SynA53T brains much earlier than in Tg α 2- α SynA53T brains”.

This figure is now quantified (**Suppl. Fig. S4f**). Samples were run on the same blot. Please see below for example:

- The authors discuss literature suggesting that the majority of extracellular aSyn is not in association with exosomes/extracellular vesicles. The Discussion would benefit from including key studies demonstrating the association of aSyn with exosomes/extracellular vesicles in the human cerebrospinal fluid, and that uptake of this fraction is capable of leading to aSyn-related pathology in recipient cells. Based on the current knowledge, it would be challenging to conclude on the relative contribution of each of the proposed pathways to aSyn release/secretion.

We apologize for the unclarity. Our study aims to detail a mechanism of α Syn aggregate release from neurons, and ruling out every non-lysosomal source, e.g. exosomes, is not the purpose of our study. Yet, we agree with this reviewer and have included (Ngolab et al., 2017) in the Introduction, and have also clarified in the Discussion that other pathways of α Syn aggregate-release cannot be excluded.

Reviewer #2 (Remarks to the Author):

The manuscript addresses the mechanism of α -synuclein secretion using *in vivo* and *in vitro* models, concluding that lysosomal exocytosis and VAMP7/ SNAP23 proteins are required for this process. The work addresses an important question in the field and uses relevant models to test their hypothesis. Most of the work is technically well done: they present clean data that shows successful isolation of endolysosomal vesicles using multiple techniques, and the rescue data with VAMP7 DN / SNAP23 KD is impressive. The nature of the secreted α -synuclein was also addressed well by *in vitro* seeding experiments.

We thank the reviewer for finding our study important and technically well done.

However, the experimental design could be improved. For example, there are a few critical aspects of the work that are conspicuously missing. This includes 1) The use of non-Tg wild-type mice to match with A53T

We have addressed this comment by showing that α Syn KO and WT neurons do not accumulate α Syn aggregates and thus do not secrete α Syn aggregates (**new Suppl. Fig S5c-d & S6**). Additionally, we show that WT mouse brains do not produce α Syn aggregates, and that WT brains and neurons have only monomeric α Syn (**Suppl. Fig. S4f and new Suppl. Fig. S5c-d**).

In contrast, Tg- α Syn^{A53T} mice overexpressing human A53T mutant synuclein develop α Syn aggregates by 6 months (**Fig. 1a, 2a, and Suppl. Fig. S4f**). It is difficult to directly compare Tg- α Syn^{A53T} mice and non-Tg WT mice, because our quantifications and (Chandra et al., 2005) show that total α Syn levels are about 8-10 fold higher in the Tg- α Syn^{A53T} mice. Therefore, we hope this reviewer will agree that pre-aggregate mice of the same Tg- α Syn^{A53T} genotype, which have only monomeric α Syn at 1 month of age (**Fig 2a**) are more accurate controls than WT mice.

2) measurements of cell death or lysis as controls, to rule them out as a source of extracellular lysosomal secretion

Actin and monomeric α Syn were shown to indicate a lack of leakage of cytoplasmic content (**Fig. 4b**). In addition, we have now included cell viability/death assays which show that cell death is not the source of extracellular α Syn species (**new Suppl. Fig. S5e-f**), and that changes in neuronal activity and cytosolic Ca²⁺, as well as VAMP7 DN or SNAP23 KD, all of which alter lysosomal exocytosis and release of α Syn species, do not affect cell death (**new Suppl. Figs. S8c-d and S9d**).

3) lack of consideration for LAMP1 localization in endosomes, which is a source of exosomes.

Lamp1 (like most lysosomal membrane proteins) will fractionally reside in many parts of the secretory system. For example, due to biogenesis it will also be in ER and Golgi stacks, and in early and late endosomes due to surface-to-lysosome trafficking. However, due to the transitional nature of these compartments, it accumulates mostly in its terminal compartment of lysosomes which have a relatively slow turnover compared to other *in-transit* secretory membranes. However, to address the reviewer's concern, we have now referred to Lamp-1 being present in late-endo/lysosomes. Importantly, we now show α Syn aggregates in lysosomes by multiple Lamp1-independent techniques: (1) Isolation of TMEM192-3xHA positive lysosomes (**new Suppl. Fig. S6**; characterized/used in (Abu-Remaileh et al., 2017; Wyant et al., 2018)), (2) visual ultrastructural evidence, using *in-situ* proximity ligation assay (**new Fig. 2c**), and (3) the previously shown dextranosome-isolation studies (**Fig 1**).

Furthermore, while the biochemistry is well done, the use of imaging techniques such as immuno-electron microscopy would go a long way in providing direct evidence for α -synuclein accumulation in lysosomes and secretion via lysosomal exocytosis.

We had previously included biochemical evidence of ultrastructural co-location by Apex-2 proximity-labeling, which labels molecules within an ultrastructural distance of 10-20nm (Hung et al., 2016; Mayer and Bendayan, 1997) (**Fig. 2b**). Now we have added visual evidence, using an *in-situ* proximity ligation assay (**new Fig. 2c**), which confirms presence of filamentous α Syn in lysosomes. Immuno-EM was forgone due to too much background/noise from >80% cytoplasmic α Syn aggregates. In addition, we show α Syn aggregates in lysosomes by other Lamp1-independent techniques, i.e. immunisolation of TMEM192-3xHA positive lysosomes (**new Suppl. Fig. S6**) plus the previously shown dextranosome-isolation studies (**Fig 1**).

The link to CSP^{-/-} and ^{+/-} was presented early in the manuscript, but then faded away without any apparent reason. It would be interesting to follow through with these studies, for example by incorporating them into the lysosomal exocytosis studies. Perhaps the phenotype would be exacerbated. Without this, the data seem a little disjointed.

This experiment was our initial indication of the involvement of lysosomes, showing a lysosome-storage defect produced by the CSP α null phenotype correlating with accumulation of α Syn aggregates. To address the specific question of whether α Syn aggregates accumulate in neuronal lysosomes, we had to deconvolute the model systems by removing the severe lysosomal storage phenotype produced by CSP α KO. We are continuing to study the CSP α knockout mouse line to understand the cellular mechanism of the CSP α mutation-linked lysosomal storage disease (adult-onset neuronal ceroid lipofuscinosis; ANCL), independent from α Syn aggregate accumulation/exocytosis.

Other specific comments are indicated below.

1) The results of figure 1 are potentially interesting, but it is a little unusual to find only aggregate forms of a-synuclein within isolated lysosomes. Monomers are known to be internalized by CMA or delivered via macroautophagy, and at least some aggregates will break down into their monomeric components during SDS-PAGE, so one would expect some detectable amounts of monomers in these fractions, or at least proteolytic fragments. Perhaps a longer exposure would bring these out, or a stronger a-synuclein antibody that also detects mouse a-synuclein, and with the total lysate loaded on the same gel. If monomers cannot be detected, then other fractions should be analyzed to show where it is, perhaps cytosol fractions.

Monomeric α Syn is a natively unfolded protein (Burré et al., 2013; Chandra et al., 2003; Kim, 1997; Uversky et al., 2001; Weinreb et al., 1996). When transported into the lysosomal lumen as an extended unfolded polypeptide (e.g. via CMA), it is likely to have a very rapid turnover from lysosomal peptidases (e.g. Figs S3b and S4a in (Cuervo et al., 2004)), which will lead to little if any steady-state presence, compared to its most degradation-resistant versions – structurally and metabolically stable amyloid type deposits that are likely to survive in the longer run. In **Fig. 1a**, α Syn monomer partitions into the cytoplasmic+light organelle fraction (labeled as “Crude cytoplasm”), reflecting its well-studied/reported subcellular localization in the cytosol and on synaptic vesicles. In contrast, α Syn aggregates partitioned with heavy organelles (lysosomes, mitochondria, and peroxisomes), which were further separated by Percoll fractionation to isolate the lysosomal fraction (**Fig. 1**). We now also show that α Syn aggregates are SDS-extractable, but are not SDS-labile to a large extent, and only partially break down with a strong chaotropic agent like urea (**new Suppl. Fig. S5c**). This experiment also shows that monomeric α Syn is mainly in the cytoplasmic/Triton X-100-soluble fractions (**new Suppl. Fig S5c**) and is not secreted (**new Suppl. Fig. S5d**).

2) Conformation of aggregated a-synuclein with lysosomes should be done by immunohistochemistry or immuno-EM.

We had previously included biochemical evidence of ultrastructural co-location by Apex-2 proximity-labeling, which labels molecules within an ultrastructural distance of 10-20nm (Hung et al., 2016; Mayer and Bendayan, 1997) (**Fig. 2b**). Now we show α Syn aggregates in lysosomes by other Lamp1-independent techniques as well: (1) visual ultrastructural evidence, using *in-situ* proximity ligation assay (**new Fig. 2c**); immuno-EM was forgone due to too much background/noise from >80% cytoplasmic α Syn aggregates, (2) immunoisolation of TMEM192-3xHA positive lysosomes (**new Suppl. Fig. S6**), (3) plus the previously shown dextranosome-isolation studies (**Fig. 1**).

3) Analysis of mice expression endogenous wt a-synuclein should be included in figure 1, including lysosomal and cytosolic fractions.

WT mouse brains and neurons do not produce α Syn aggregates (**Suppl. Fig. S4f and new Suppl. Fig. S5c**) but have only monomeric α Syn. Tg- α Syn^{A53T} mice overexpressing human A53T mutant synuclein develop α Syn aggregates by 6 months (**Figs. 1a, 2a, and Suppl. Fig. S4f**), hence the age comparison between 1 (pre-aggregate control) and 6-month-old mice. Importantly, monomeric α Syn (even when overexpressed in transgenic mice) is not present in the heavy

organelle fractions (comprising lysosomes, mitochondria, and peroxisomes), but only in the crude cytoplasmic fractions which includes the cytosol and light membranes such as synaptic vesicles (**Fig. 1a**).

Total α Syn levels are about 8-10 fold higher in Tg- α Syn^{A53T} mice compared to non-Tg WT mice (our quantifications and (Chandra et al., 2005)), making direct comparisons between the two mouse lines less useful. Therefore, we submit that pre-aggregate Tg- α Syn^{A53T} mice of the same genotype (which have only monomeric α Syn at 1 month (**Fig 2a**)) are less variable and more accurate controls than WT mice.

4) I am not sure that I understand why or how dead mice could have been included in the behavioral test data of Figure S3f/g, S4d/e. Perhaps other behavioral tests could provide be a better readout. It would be important to know if overexpression of LAMP1 influenced the baseline A53T genotype.

We do show that presence of Lamp1 transgene does not influence the baseline A53T phenotype (**Suppl. Fig. S3 red vs. purple**) as seen by survival, hind-limb clamping, and motor behavioral data (grid-hang, pole-turn, and pole descent). The progressive neurodegenerative phenotype of the A53T mice results in their death which makes analysis of behavioral data scored numerically difficult. Categorical analysis of the behavior data (binary yes or no conditions) would merely precede and mirror the survival data. This is seen in the hand-limb clamping data (yes or no clasp) and grid-hang data in which the mice were scored based on if they can hang on for more than 120 seconds. We agree that there needs to be better ways to depict data to account for mouse mortality. Some papers show bar graphs comparing groups at the last timepoint, but by omitting data, the progressive deterioration of the mice is lost.

If the sick mouse is deleted from the remaining cohort after its death, the cohort becomes healthier in the next measurement, which is untrue and misleading. This type of curve shows wrongly the cohort becoming sick only when the last few mice are left, i.e. when all or most of them get sick at the same time. Until then, the phenotype keeps bouncing to perfect behavioral-health after each death. This problem appears to be unsolved in statistical analyses even in clinical studies with high mortality (Colantuoni et al., 2018). Thus, once a mouse died, we gave it the worst score for the assay for the remaining trials, retaining its status as "hit the worst score". For instance, the worst (maximum) score for pole turn and pole descent is set at 20 seconds, after which we consider the mouse fully disabled in this assay. For transparency, this was clearly mentioned in the legends and the graphs where mice were removed once they had died (darker shaded lines) were shown in each panel (**Suppl. Figs. S3f-g, and S4d-e**).

5) There is no indication of how many times the new LAMP1 -A53T mice were backcrossed prior to doing experiments to assure that each of the 4 lines are comparable in terms of their genetic background.

Thank you for pointing this out. We have now added this information in the Methods section under the mouse lines used.

6) In figure 2, the 1 month control is informative, but it will be essential to include a LAMP1 Tg expressing wild-type endogenous mouse α -synuclein in this experiment.

We have shown that Lamp1 transgenics (expressing WT endogenous mouse α Syn) are identical health-wise to their WT littermates (**old Figs. S2 and S3**). We also show that WT mouse brains and neurons do not have aggregates (**Suppl. Fig. S4f and new Suppl. Fig. S5c**), and that WT neurons do not secrete monomeric α Syn into the medium (**new Suppl. Fig. S5d**).

7) Neuronal health should be assessed in primary culture experiments of DIV 14 and 49, to determine if toxicity occurs during the long term culture. LAMP1 or another lysosomal marker in addition to CathL should be included.

Actin and monomeric α Syn were shown to indicate a lack of leakage of cytoplasmic material (**Fig. 4b**). In addition, we have now included cell viability/death assays (**new Suppl. Figs. S5e-f**) and the finding that changes in secretion of α Syn aggregates into the medium do not correlate with cell viability or death (**new Suppl. Figs. S8d and S9d**).

8) Figure S6 is interesting, however it requires WT and non-aged tg controls. It would be critical to assess contamination by other material such as blood cells etc to make sure the signal is derived only from CSF.

WT and non-aged tg mice do not develop α Syn aggregates (**Suppl. Fig. S4f**). We have now included proteolysis experiment with and without addition of the detergent Triton X-100 from the culture medium (**new Fig. 3b**), where contamination from blood cells is not a concern.

9) Cell lysis or death needs to be excluded as a potential source of extracellular a-synuclein.

Actin and monomeric α Syn were shown to indicate a lack of leakage of cytoplasmic material (**Fig. 4b**). In addition, we have now included cell viability/death assays (**new Suppl. Figs. S5e-f**), and the finding that changes in secretion of α Syn aggregates into the medium do not correlate with cell viability or death (**new Suppl. Figs. S8c-d and S9d**).

10) LAMP1 does not exclusively label lysosomes but also late endosomes and a small portion of early endosomes (Saftig and Klumperman, 2009). The endosomal system feeds vesicles into multivesicular bodies (MVBs), which are the main source of exosomes (Trajkovic et al Science 2008). Therefore it is possible that the source of a-synuclein is from exosomes. It also somewhat surprising that tsg101 isn't biotinylated by a LAMP1 conjugate, given its clear association with the endosomal system. Perhaps tsg101 is not exposed to the lumen. More direct experiments would be required prior to ruling out an exosomal source of a-synuclein.

Lamp1 (like most lysosomal membrane proteins) will fractionally reside in many parts of the secretory system. For example, due to biogenesis it will also be in ER and Golgi stacks, and in early and late endosomes due to surface-to-lysosome trafficking. However, due to the transitional nature of these compartments, it accumulates mostly in its terminal compartment of lysosomes which have a relatively slow turnover compared to other in-transit secretory membranes. However, to address the reviewer's concern, we have now referred to Lamp-1 being present in late-endo/lysosomes.

Importantly, we now show α Syn aggregates in lysosomes by multiple Lamp1-independent techniques: (1) Isolation of TMEM192-3xHA positive lysosomes (**new Suppl. Fig. S6**; characterized/used in (Abu-Remaileh et al., 2017; Wyant et al., 2018)), (2) visual ultrastructural evidence, using *in-situ* proximity ligation assay (**new Fig. 2c**), and (3) the previously shown dextranosome-isolation studies (**Fig 1**).

Proteolysis experiments, from CSF (**Fig. 3a**) and now from culture medium (**Fig. 3b**) also argue against exosome-enveloped aggregates as the main form of aggregates. However, ruling out exosomes and/or every non-lysosomal source is not the purpose of this study. Thus, we have now included a statement that these experiments do not rule out other possible sources.

We agree that the reviewer's topological reasoning for the lack of TSG101 labeling is the most likely one.

11) Throughout the manuscript, the MW of Cath L should be included, as it is composed of heavy and light chains. Determination of Cath L enzyme activity or other hydrolases in the media would strengthen the conclusions that the media contents are derived from lysosomes.

We have now indicated in the figures that mature Cathepsin-L (Cath-L^{mat}) is shown (~27kDa); after propeptide removal, any further cleavage(s) of Cath-L^{mat} (e.g. into heavy and light chains) is not as clear as with Cath-D (Mason et al., 1985; Mason et al., 1984). Importantly, the Cath-L^{mat} band shown correlates directly with Cath-D activity (**Fig. 1**).

12) In Figure 3B, it is possible that VAMP7 DN or SNAP23 KD are improving cell health, resulting in less a-synuclein in the media. This possibility could be easily ruled out with a cell death assay.

Actin and monomeric α Syn were shown to indicate a lack of leakage of cytoplasmic material (**Fig. 4b**). In addition, we have now included cell viability/death assays which show that changes in secretion of α Syn aggregates into the medium do not correlate with cell viability/death (**new Suppl. Figs. S5e-f**), and we also show that reduced secretion of α Syn aggregates due to SNAP-23 KD and Vamp7 DN are not caused by cytoprotection (**new Suppl. Fig. S9d**).

13) It would be important to know if WT neurons secrete any a-synuclein, since the media was used as a control in figure

4A, B. Immunodepleting α -synuclein from the media would be an important control here. If amyloidogenic seeds are formed and secreted, then sample agitation should not be required to stimulate recombinant myc- α -synuclein aggregation.

WT mouse brains and neurons do not develop α Syn aggregates (**Fig. S4f, new Suppl. Fig. S5c**), thus there are no α Syn aggregates or monomers secreted in the medium of WT neurons (**new Suppl. Fig. S5d**). Even when α Syn is overexpressed in transgenic mice, we do not find monomeric α Syn or other natively-cytosolic proteins (e.g. actin) in the medium (**Fig. 4b**).

Agitation helps to further the cycles of aggregate-assembly/propagation after the seeds initiate. Importantly, it is done identically to all samples, and is much more neutral compared to the addition of chemical accelerants (e.g. heparin) which themselves may template aggregate-assembly.

Reviewer #3 (Remarks to the Author):

This study does a nice job of documenting the lysosomal localization of pathogenic forms of α Syn, which was proposed previously by several groups. The original part of the study is the evidence indicating that SNARE-mediated lysosomal exocytosis is involved in the release of α Syn aggregates.

We thank the reviewer for the positive evaluation of our manuscript.

However, lysosomal exocytosis has been extensively demonstrated in other cell types to be absolutely dependent on calcium transients in the donor cells, and the authors do not even discuss this. Is calcium the trigger?

We have now included the effect of neuronal activity and cytosolic calcium on the lysosomal exocytosis of α Syn aggregates (**new Suppl. Fig. S8**), and find that cytosolic calcium and neuronal activity do enhance lysosomal exocytosis of α Syn aggregates in neurons. The physiological regulation of lysosomal exocytosis is a fascinating subject and should be explored further in future studies.

As presented, the study is incomplete because it also does not clarify how the exocytosis of α Syn aggregates leads to cell-cell transfer.

Cell-to-cell transfer of α Syn aggregates is a multistep process. Our manuscript describes the release mechanism of α Syn aggregates from neurons. We agree that the subsequent processes, e.g. potential ways of clearance by glia, glymphatics, and/or other processes, and the potential uptake mechanism(s) of the released aggregates into the next cell(s) by glial cells and/or neurons, and how exactly those cells deal with internalized aggregates, are all fascinating and important topics that many labs are trying to piece together. However, we submit that all these processes do not need to be clarified in this study to understand the release-mechanism of aggregated α Syn, which in turn forms the basis for spread. We have now added an experiment showing that the aggregates released via lysosomal exocytosis are capable of seeding aggregation not only of purified α Syn (**Fig. 5a-c**), but also of α Syn in neurons (**new Fig. 5d**), suggesting that they template not just purified, but neuronally expressed α Syn.

Main points:

1) In the introduction, the authors refer to Abounit et al. EMBO J 2016 as describing the role of tunneling nanotubes on trafficking of α Syn aggregates. However, they omit something important in their statement: the Abounit paper showed that that >90% of α Syn fibrils co-localized with lysosomal markers, and proposed that lysosomes were the organelles transferred from cell-to-cell through nanotubes. There seem to be additional studies showing that α Syn accumulates in lysosomes (Sung 2001, Lee 2008, Liu 2009, Hasegawa 2011). Thus, this statement in the introduction is somewhat misleading: "we present our finding that pathogenic α Syn species accumulate within the neuronal lysosomes in mouse brains and in primary neurons". The authors may have demonstrated more precisely the lysosomal localization using fractionation and proximity labeling methods, with good distinction between monomers and pathogenic species - but they

must make clear which of their results are confirmatory of previous studies, and which are absolutely novel (such as release through lysosomal exocytosis).

We have modified the statement to reflect what we exactly show in the study, instead of stating “we present our findings”. Previous work including the papers cited by the reviewer (Hasegawa et al., 2011; Lee et al., 2008; Liu et al., 2009a; Sung et al., 2001) do not address lysosomes as the accumulation-site for neuronally produced α Syn aggregates, nor identify lysosomal exocytosis as a potential source of extracellular α Syn aggregates. These studies mainly address the uptake of externally fed α Syn (Sung et al., 2001) or α Syn fibrils (Lee et al., 2008; Liu et al., 2009a) taken up by cells, while our findings are based on accumulation within lysosomes of the neuronally produced α Syn aggregates. Furthermore, in some studies lysosomal protease inhibition is used to produce α Syn accumulation in lysosomes (Hasegawa et al., 2011; Lee et al., 2008; Liu et al., 2009b). But overall, these papers do not speculate beyond seeing lysosomes as a degradative organelle. In addition, our results focus on the release of α Syn aggregates into the extracellular space.

Now we refer to the (Abounit et al., 2016; Dilsizoglu Senol et al., 2021) studies for showing lysosomal movement through tunneling nanotubes, in agreement with the reviewer, and apologize for the unclarity of our previous statement. We have also added in the discussion that our results do not exclude other pathways of α Syn aggregate release.

2) In Fig. 3a, which fraction of the total pathogenic α Syn species present in the cells was released into the supernatant? Is this fraction similar for Cath-L and α Syn?

We performed this as a yes/no (detectable vs. not) experiment (**now in Fig. 4a**), mainly because of factors such as biotinylation efficiency and capture efficiency: (i) Numbers of biotinylation-accessible Tyr side chains on cathepsin-L vs. α Syn, (ii) accessibility of biotinylated sites to streptavidin due to different folds of cathepsin-L vs. α Syn, and (iii) Streptavidin-accessible biotinylated site(s) on one/few α Syn molecules allowing capture of the unbiotinylated molecules associated in the aggregate as well (and then its detection in western blot). Despite these sources of variation, as shown below, the streptavidin-captured levels of α Syn and cathepsin-L are not very different. If the reviewer wishes we can add this quantification to the paper, but due to the reasons above, it would not hold as much meaning as the quantifications and correlations shown in **Figs. 4b-c** – which show that the levels of α Syn aggregates released is correlated to the levels of Cathepsin-L released into the media through the various SNARE manipulations.

Figure 4a Quantitation

(a) Tg^{x2}- α Syn^{A53T} primary neurons lentivirally expressing a synapsin-1 promoter-driven A_{pex-2} Lamp1 (infected on DIV 7) were subjected to proximity-labeling on DIV 47. Biotinylated proteins released into the media during 48 h (by DIV 49) were precipitated on streptavidin-magnetic beads and immunoblotted for the indicated α Syn species (α Syn, α Syn^{pSer129}, α Syn^{Amyl}, and α Syn^{Fila}), lysosome luminal protein cathepsin-L, and exosome luminal protein TSG101. Strept. PPT. = streptavidin precipitate. (representative of n=3). For quantifications shown here, amount of protein precipitated from the medium was normalized to its levels in the total medium (5% loaded for immunoblots and 10% loaded for dot blots)

(n=3). All data represent means \pm SEM. Each 'n' corresponds to a batch of neuronal culture and infection. In (a) n.s. = not significant; *P<0.05; **P<0.01; ***P<0.001 by two-tailed Student's t-test.

3) The authors cite Rao et al JBC in support of the involvement of VAMP7 and SNAP23 in lysosomal exocytosis. However, they omit something important – that study involved calcium-triggered lysosomal exocytosis. What is the role of calcium in their system? This is critical because only very low levels of lysosomal content release is observed, in many cell types, in the absence of calcium influx. They make a distinction on page 4 between lysosomal exocytosis of cathepsin L and constitutive exocytosis of neuroserpin, but do not explain what they mean by this. If not constitutive, is lysosomal exocytosis and pathological aSyn release calcium dependent? Can exocytosis be increased by inducing calcium transients in the cells, and blocked by calcium chelators?

Neuroserpin release is termed constitutive simply because it occurs by the conventional (and non-lysosomal) protein secretion pathway; we have now clarified this in the text. Lysosomal exocytosis in general does appear to be modulated by Ca²⁺, in agreement with the Andrews lab papers, including Rao et al. JBC. We have now included effects of neuronal activity and cytosolic calcium on the lysosomal exocytosis of α Syn aggregates (**new Suppl. Fig. S8**).

4) The in vitro seeding of aSyn aggregates by supernatants containing secreted pathogenic aSyn is intriguing, but does not explain at all how these aggregates enter other cells. Are they taken up by endocytosis or phagocytosis? If they are, this would explain how they arrive in lysosomes, without a need to invoke autophagy. Without clarifying this aspect, the study is incomplete and does not advance the field much beyond previous publications.

This particular experiment (**Fig. 5a-c**) is included to specifically show that the α Syn aggregate species released by lysosomal exocytosis (thus reduced by Vamp7 DN) are capable of templating purified α Syn into aggregates. We have now added experiments showing that the aggregates released via lysosomal exocytosis are capable of seeding/accelerating aggregation in neurons (**new Fig. 5d**), i.e. they template not just purified, but neuronally expressed α Syn.

We agree that the subsequent processes, e.g. potential ways of clearance by glia, glymphatics, and/or other processes, and the potential uptake mechanism(s) of the released aggregates into the next cell(s) by glial cells and/or neurons, and how exactly those cells deal with internalized aggregates, are all fascinating and important topics that many groups are working on. However, we submit that all these processes do not need to be clarified in this study to understand the release-mechanism of aggregated α Syn.

Minor point:

The Ejlerskov 2013a and 2013b references are exactly the same.

Thanks for noting. This is fixed now.

References for response to reviewers

- Abounit, S., Bousset, L., Loria, F., Zhu, S., de Chaumont, F., Pieri, L., Olivo-Marin, J.C., Melki, R., and Zurzolo, C. (2016). Tunneling nanotubes spread fibrillar α -synuclein by intercellular trafficking of lysosomes. *The EMBO journal* **35**, 2120-2138.
- Abu-Remaileh, M., Wyant, G.A., Kim, C., Laqtom, N.N., Abbasi, M., Chan, S.H., Freinkman, E., and Sabatini, D.M. (2017). Lysosomal metabolomics reveals V-ATPase- and mTOR-dependent regulation of amino acid efflux from lysosomes. *Science* **358**, 807-813.
- Anderson, J.P., Walker, D.E., Goldstein, J.M., de Laat, R., Banducci, K., Caccavello, R.J., Barbour, R., Huang, J., Kling, K., Lee, M., *et al.* (2006). Phosphorylation of Ser-129 is the dominant pathological modification of alpha-synuclein in familial and sporadic Lewy body disease. *J Biol Chem* **281**, 29739-29752.
- Bagchi, S., Fredriksson, R., and Wallén-Mackenzie, Å. (2015). In Situ Proximity Ligation Assay (PLA). *Methods Mol Biol* **1318**, 149-159.
- Burré, J., Vivona, S., Diao, J., Sharma, M., Brunger, A.T., and Südhof, T.C. (2013). Properties of native brain α -synuclein. *Nature* **498**, E4-6; discussion E6-7.
- Chandra, S., Chen, X., Rizo, J., Jahn, R., and Südhof, T.C. (2003). A broken alpha-helix in folded alpha-Synuclein. *J Biol Chem* **278**, 15313-15318.
- Chandra, S., Gallardo, G., Fernández-Chacón, R., Schlüter, O.M., and Südhof, T.C. (2005). Alpha-synuclein cooperates with CSPalpha in preventing neurodegeneration. *Cell* **123**, 383-396.
- Colantuoni, E., Scharfstein, D.O., Wang, C., Hashem, M.D., Leroux, A., Needham, D.M., and Girard, T.D. (2018). Statistical methods to compare functional outcomes in randomized controlled trials with high mortality. *BMJ* **360**, j5748.
- Cuervo, A.M., Stefanis, L., Fredenburg, R., Lansbury, P.T., and Sulzer, D. (2004). Impaired degradation of mutant alpha-synuclein by chaperone-mediated autophagy. *Science* **305**, 1292-1295.
- Dilsizoglu Senol, A., Samarani, M., Syan, S., Guardia, C.M., Nonaka, T., Liv, N., Latour-Lambert, P., Hasegawa, M., Klumperman, J., Bonifacino, J.S., *et al.* (2021). α -Synuclein fibrils subvert lysosome structure and function for the propagation of protein misfolding between cells through tunneling nanotubes. *PLoS Biol* **19**, e3001287.
- Elfarrash, S., Jensen, N.M., Ferreira, N., Betzer, C., Thevathasan, J.V., Diekmann, R., Adel, M., Omar, N.M., Boraie, M.Z., Gad, S., *et al.* (2019). Organotypic slice culture model demonstrates inter-neuronal spreading of alpha-synuclein aggregates. *Acta Neuropathologica Communications* **7**, 213.
- Fontaine, S.N., Zheng, D., Sabbagh, J.J., Martin, M.D., Chaput, D., Darling, A., Trotter, J.H., Stothert, A.R., Nordhues, B.A., Lussier, A., *et al.* (2016). DnaJ/Hsc70 chaperone complexes control the extracellular release of neurodegenerative-associated proteins. *Embo j* **35**, 1537-1549.
- Fujiwara, H., Hasegawa, M., Dohmae, N., Kawashima, A., Masliah, E., Goldberg, M.S., Shen, J., Takio, K., and Iwatsubo, T. (2002). α -Synuclein is phosphorylated in synucleinopathy lesions. *Nature Cell Biology* **4**, 160-164.
- Fusco, G., Chen, S.W., Williamson, P.T.F., Cascella, R., Perni, M., Jarvis, J.A., Cecchi, C., Vendruscolo, M., Chiti, F., Cremades, N., *et al.* (2017). Structural basis of membrane disruption and cellular toxicity by α -synuclein oligomers. *Science* **358**, 1440-1443.
- Guo, M., Wang, J., Zhao, Y., Feng, Y., Han, S., Dong, Q., Cui, M., and Tieu, K. (2020). Microglial exosomes facilitate α -synuclein transmission in Parkinson's disease. *Brain* **143**, 1476-1497.
- Hasegawa, T., Konno, M., Baba, T., Sugeno, N., Kikuchi, A., Kobayashi, M., Miura, E., Tanaka, N., Tamai, K., Furukawa, K., *et al.* (2011). The AAA-ATPase VPS4 regulates extracellular secretion and lysosomal targeting of α -synuclein. *PLoS One* **6**, e29460.
- Hung, V., Udeshi, N.D., Lam, S.S., Loh, K.H., Cox, K.J., Pedram, K., Carr, S.A., and Ting, A.Y. (2016). Spatially resolved proteomic mapping in living cells with the engineered peroxidase APEX2. *Nat Protoc* **11**, 456-475.
- Kim, J. (1997). Evidence that the precursor protein of non-A beta component of Alzheimer's disease amyloid (NACP) has an extended structure primarily composed of random-coil. *Molecules and cells* **7**, 78-83.
- Kim, S., Kwon, S.H., Kam, T.I., Panicker, N., Karuppagounder, S.S., Lee, S., Lee, J.H., Kim, W.R., Kook, M., Foss, C.A., *et al.* (2019). Transneuronal Propagation of Pathologic α -Synuclein from the Gut to the Brain Models Parkinson's Disease. *Neuron* **103**, 627-641.e627.

- Kornilov, R., Puhka, M., Mannerström, B., Hiidenmaa, H., Peltoniemi, H., Siljander, P., Seppänen-Kajjansinkko, R., and Kaur, S. (2018). Efficient ultrafiltration-based protocol to deplete extracellular vesicles from fetal bovine serum. *J Extracell Vesicles* 7, 1422674-1422674.
- Lassen, L.B., Gregersen, E., Isager, A.K., Betzer, C., Kofoed, R.H., and Jensen, P.H. (2018). ELISA method to detect α -synuclein oligomers in cell and animal models. *PLoS One* 13, e0196056.
- Lau, A., So, R.W.L., Lau, H.H.C., Sang, J.C., Ruiz-Riquelme, A., Fleck, S.C., Stuart, E., Menon, S., Visanji, N.P., Meisl, G., *et al.* (2020). α -Synuclein strains target distinct brain regions and cell types. *Nature neuroscience* 23, 21-31.
- Lee, H.J., Suk, J.E., Bae, E.J., Lee, J.H., Paik, S.R., and Lee, S.J. (2008). Assembly-dependent endocytosis and clearance of extracellular alpha-synuclein. *The international journal of biochemistry & cell biology* 40, 1835-1849.
- Lindersson, E., Beedholm, R., Højrup, P., Moos, T., Gai, W., Hendil, K.B., and Jensen, P.H. (2004). Proteasomal Inhibition by α -Synuclein Filaments and Oligomers*. *Journal of Biological Chemistry* 279, 12924-12934.
- Liu, J., Zhang, J.P., Shi, M., Quinn, T., Bradner, J., Beyer, R., Chen, S., and Zhang, J. (2009a). Rab11a and HSP90 regulate recycling of extracellular alpha-synuclein. *J Neurosci* 29, 1480-1485.
- Liu, Z., Meray, R.K., Grammatopoulos, T.N., Fredenburg, R.A., Cookson, M.R., Liu, Y., Logan, T., and Lansbury, P.T., Jr. (2009b). Membrane-associated farnesylated UCH-L1 promotes alpha-synuclein neurotoxicity and is a therapeutic target for Parkinson's disease. *Proc Natl Acad Sci U S A* 106, 4635-4640.
- Luk, K.C., Kehm, V., Carroll, J., Zhang, B., O'Brien, P., Trojanowski, J.Q., and Lee, V.M. (2012). Pathological α -synuclein transmission initiates Parkinson-like neurodegeneration in nontransgenic mice. *Science* 338, 949-953.
- Mao, X., Ou, M.T., Karuppagounder, S.S., Kam, T.I., Yin, X., Xiong, Y., Ge, P., Umanah, G.E., Brahmachari, S., Shin, J.H., *et al.* (2016). Pathological α -synuclein transmission initiated by binding lymphocyte-activation gene 3. *Science* 353.
- Mason, R.W., Green, G.D., and Barrett, A.J. (1985). Human liver cathepsin L. *Biochem J* 226, 233-241.
- Mason, R.W., Taylor, M.A., and Etherington, D.J. (1984). The purification and properties of cathepsin L from rabbit liver. *The Biochemical journal* 217, 209-217.
- Mayer, G., and Bendayan, M. (1997). Biotinyl-tyramide: a novel approach for electron microscopic immunocytochemistry. *J Histochem Cytochem* 45, 1449-1454.
- Ngolab, J., Trinh, I., Rockenstein, E., Mante, M., Florio, J., Trejo, M., Masliah, D., Adame, A., Masliah, E., and Rissman, R.A. (2017). Brain-derived exosomes from dementia with Lewy bodies propagate alpha-synuclein pathology. *Acta Neuropathol Commun* 5, 46.
- Nuber, S., Harmuth, F., Kohl, Z., Adame, A., Trejo, M., Schönig, K., Zimmermann, F., Bauer, C., Casadei, N., Giel, C., *et al.* (2013). A progressive dopaminergic phenotype associated with neurotoxic conversion of α -synuclein in BAC-transgenic rats. *Brain* 136, 412-432.
- Paleologou, K.E., Kragh, C.L., Mann, D.M.A., Salem, S.A., Al-Shami, R., Allsop, D., Hassan, A.H., Jensen, P.H., and El-Agnaf, O.M.A. (2009). Detection of elevated levels of soluble α -synuclein oligomers in post-mortem brain extracts from patients with dementia with Lewy bodies. *Brain* 132, 1093-1101.
- Sung, J.Y., Kim, J., Paik, S.R., Park, J.H., Ahn, Y.S., and Chung, K.C. (2001). Induction of neuronal cell death by Rab5A-dependent endocytosis of alpha-synuclein. *J Biol Chem* 276, 27441-27448.
- Uversky, V.N., Lee, H.J., Li, J., Fink, A.L., and Lee, S.J. (2001). Stabilization of partially folded conformation during alpha-synuclein oligomerization in both purified and cytosolic preparations. *J Biol Chem* 276, 43495-43498.
- Volpicelli-Daley, L.A., Luk, K.C., Patel, T.P., Tanik, S.A., Riddle, D.M., Stieber, A., Meaney, D.F., Trojanowski, J.Q., and Lee, V.M. (2011). Exogenous α -synuclein fibrils induce Lewy body pathology leading to synaptic dysfunction and neuron death. *Neuron* 72, 57-71.
- Weinreb, P.H., Zhen, W., Poon, A.W., Conway, K.A., and Lansbury, P.T., Jr. (1996). NACP, a protein implicated in Alzheimer's disease and learning, is natively unfolded. *Biochemistry* 35, 13709-13715.
- Wyant, G.A., Abu-Remaileh, M., Frenkel, E.M., Laqtom, N.N., Dharamdasani, V., Lewis, C.A., Chan, S.H., Heinze, I., Ori, A., and Sabatini, D.M. (2018). NUFIP1 is a ribosome receptor for starvation-induced ribophagy. *Science* 360, 751-758.

REVIEWER COMMENTS

Reviewer #1 (Remarks to the Author):

In the revised study, the authors have included new experiments, further strengthening the evidence that pathogenic alpha synuclein species associate with neuronal endolysosomes, and released by calcium-regulated SNARE-dependent lysosomal exocytosis. The following points remain to be addressed:

- **Fig. 2a, corresponding to the immuno-isolation of neuronal endolysosomes from the transgenic Tg- α SynA53T/Tg-HALamp1Myc mouse brain presents one of the key experiments of the study. The model is new, thus in addition to the western blot, it is relevant to include immunofluorescence micrographs of mouse cerebral tissue at 1 and 6 months of age, showing the neuronal expression of the transgene throughout different brain regions.**

- **Fig. S2a and S7b: For visual clarity to the reader, it would be good to include individual as well as the merged channel, and to improve the contrast. Similarly, contrast enhancement is needed for Fig. S5b.**

- **In the Materials and Methods, some sections need additional details, including:**

Total number of mice/collected total CSF volume necessary for experiments in Fig. 3a.

Starting material (number of cells/protein amount) and buffer volumes used in the serial protein extraction of primary neurons.

For endolysosome immuno-isolation and SNARE protein co-IP experiments, please include more information on the protein amount or volume of tissue/neuron homogenate mixed with respective IP antibody. Also, include an estimation of the protein loaded for the corresponding western blots.

MOI for lentiviral transduction in the different cell culture experiments.

In Materials pg. 34 of 52, the section indicates endolysosome immunoprecipitation from Tg-HALamp1Myc mouse brains. Also Tg- α SynA53T/HALamp1Myc mice should be added, corresponding to Fig. 2a.

Reviewer #2 (Remarks to the Author):

The authors have addressed some of the concerns, but significant issues remain relating to the overall rigor and lack of critical controls, which reduce enthusiasm to support publication. Some of the controls that were requested were not included, leaving the work underdeveloped and of questionable relevance. There is also a novelty issue in terms of the conceptual advance. Specific points are below.

- 1) **There was a request to show that aggregated α -synuclein is in lysosomes using imaging techniques, specifically ultrastructural (Immuno-electron microscopy). The reason for this request is, while the fractionation itself appears clean, the biochemical data showing α -synuclein smears or dot blots in vivo is weak. None of the dot blots throughout the work have a membrane-matched loading control, which makes the data of limited value.**

Electron microscopy (EM), or high resolution confocal, is required to directly visualize α -synuclein aggregates within lysosomes. Unfortunately, PLA or biochemistry do not

provide ultrastructural evidence regardless of the nm distance detection that PLA provides. One confusing issue is how the authors use the term 'ultrastructure'. The term should be removed because it is misleading, unless they can provide actual EM evidence. In the absence of stronger imaging data, stating that a-synuclein "accumulates within the lumen of neuronal lysosomes" in the model is not supported by the data. A typical bilayer is ~ 4-5nm in width, and PLA cannot determine if aggregates are inside the lysosome, on surface, or adjacent in the cytosol. The quality and resolution of the PLA data can also be improved.

2) There was a request to examine endogenous wt a-synuclein. Perhaps the reasoning for this request was not clear. While it is obvious that WT mice do not have a-synuclein aggregates, the evidence that this pathway is specific for exocytosis of aggregated a-synuclein and is involved in disease pathogenesis is lacking. It is well established that a-synuclein monomers are processed through lysosomes, and that exocytosis of physiological forms occurs in primary cultures (eg Lee et al J Neurosci 2005). More importantly, there is a concern that the artificial overexpression by the TG mice, which is very unnatural, causes irrelevant phenotypes. The release of a-synuclein could occur simply because there is too much protein expressed. Showing an n=1 lyso-IP of 1 month A53T in Fig 2A, and DIV 21 in Fig S6 with no quantifications or evidence or reproducibility, are very weak controls and do not alleviate this concern. There are several additional missing specificity controls that are required. For example, does dramatic overexpression of an unrelated lysosomal substrate also lead to its exocytosis? Do A53T mice exhibit increased lysosomal exocytosis of other substrates, compared to non-Tg mice? Does endogenous a-synuclein (aggregated or not) travel through the same pathway? It seems that the authors have a good system to address the fate of endogenous wt a-synuclein that accumulates in the CSP-/- model, but this was not attempted. Without these data, the relevance and novelty of these studies is limited.

3) In vitro aggregation assays in Figure 5 remain poorly controlled. There is no molecular crowding control, and no a-synuclein- immunodepleted control. The use of SNCA -/- media doesn't alleviate this concern, since other non- a-synuclein substrates also likely accumulate and are released from lysosomes in this model, and could promote aggregation through cross seeding, molecular crowding or other mechanisms. If seeds are truly present, agitation is not needed (Buell et al PNAS 2014), and other chemical additives are would certainly confound the results further.

4) In figure 5C, the authors inappropriately equate the HMW bands with filamentous amyloid-type aggregates. There is no evidence to support these structural statements from a western blot analysis.

5) There is a novelty issue regarding the conceptual advance. Much work on a-synuclein in lysosomes / transfer has been explored, and recent publications already have described a mechanistic role of SNAP23 and VAMP7 in the exocytosis of a-synuclein (Zhao et al, Mol Neurobio 2021). Lysosomal accumulation of pathogenic a-synuclein has been shown in multiple studies including Lee HJ et al J Neurosci 2005, and Senol AD et al PLOS Bio 2021 both showing direct ultrastructural evidence for a-synuclein inside lysosome. Lysosomal exocytosis has been previously shown as a release mechanism, some with overexpression of a-synuclein, and others with endogenous and more relevant WT expression (Zhao et al 2021; Senol AD, et al 2021; Tsunemi et al J. Neurosci 2019).

The intro should include a better description of the link between a-synuclein and lysosomes. The authors indicate that the csp-/- a-synuclein data in the first part of the results 'provided a clue' that a-synuclein could accumulate in lysosomes, when in fact there is already a large amount of published data suggesting this. For example, synuclein is a common pathology seen in many lysosomal disorders (Shachar et al Mov. Dis. 2011). Multiple experimental studies have shown links between synuclein and lysosomes.

Reviewer #3 (Remarks to the Author):

The authors have adequately addressed all my comments, and this revised version is significantly improved. I have no further concerns.

Reviewer #1 (Remarks to the Author):

In the revised study, the authors have included new experiments, further strengthening the evidence that pathogenic alpha synuclein species associate with neuronal endolysosomes, and released by calcium-regulated SNARE-dependent lysosomal exocytosis. The following points remain to be addressed:

We thank the reviewer for finding that our new experiments have further strengthened the evidence. We also thank the reviewer for a careful and thorough review, which has improved the depth of this paper.

• Fig. 2a, corresponding to the immuno-isolation of neuronal endolysosomes from the transgenic Tg- α SynA53T/Tg-HALamp1Myc mouse brain presents one of the key experiments of the study. The model is new, thus in addition to the western blot, it is relevant to include immunofluorescence micrographs of mouse cerebral tissue at 1 and 6 months of age, showing the neuronal expression of the transgene throughout different brain regions.

For our new mouse line Tg-^HAALamp1^{Myc}, we have now included immunofluorescence studies of mouse brain tissues at 1 and 6 months of age, showing the neuronal expression of the transgene in cortex and hippocampus (**New Fig. 2d**).

• Fig. S2a and S7b: For visual clarity to the reader, it would be good to include individual as well as the merged channel, and to improve the contrast. Similarly, contrast enhancement is needed for Fig. S5b.

We have now added the requested individual channels in Suppl. Figs. S2a (now the new Suppl. Fig. S2b-c) and in Suppl. Fig. S7b. We have also enhanced the contrast in Suppl. Fig. S5b.

• In the Materials and Methods, some sections need additional details, including:

Total number of mice/collected total CSF volume necessary for experiments in Fig. 3a.

We have now added this information in the Methods section.

Starting material (number of cells/protein amount) and buffer volumes used in the serial protein extraction of primary neurons.

We have now added the total protein amount and buffer volumes in the starting material to the Methods section.

For endolysosome immuno-isolation and SNARE protein co-IP experiments, please include more information on the protein amount or volume of tissue/neuron homogenate mixed with respective IP antibody. Also, include an estimation of the protein loaded for the corresponding western blots.

We have now added the requested information in the Methods sections describing endolysosome immuno-isolation and SNARE protein co-IP experiments.

MOI for lentiviral transduction in the different cell culture experiments.

Thanks for the suggestion. We have now included this in the Methods.

In Materials pg. 34 of 52, the section indicates endolysosome immunoprecipitation from Tg-HALamp1Myc mouse brains. Also Tg- α SynA53T/HALamp1Myc mice should be added, corresponding to Fig. 2a.

We agree and have now included both mouse lines in the Methods section.

Reviewer #2 (Remarks to the Author):

The authors have addressed some of the concerns, but significant issues remain relating to the overall rigor and lack of critical controls, which reduce enthusiasm to support publication. Some of the controls that were requested were not included, leaving the work underdeveloped and of questionable relevance. There is also a novelty issue in terms of the conceptual advance. Specific points are below.

We thank the reviewer for spending her/his valuable time to review our work. We have worked hard to address the points raised, including providing data to demonstrate presence of α Syn aggregates inside lysosomal lumen via EM and limited proteolysis of immunocaptured lysosomes.

1) There was a request to show that aggregated a-synuclein is in lysosomes using imaging techniques, specifically ultrastructural (Immuno-electron microscopy). The reason for this request is, while the fractionation itself appears clean, the biochemical data showing a-synuclein smears or dot blots *in vivo* is weak. None of the dot blots throughout the work have a membrane-matched loading control, which makes the data of limited value. Electron microscopy (EM), or high resolution confocal, is required to directly visualize a-synuclein aggregates within lysosomes. Unfortunately, PLA or biochemistry do not provide ultrastructural evidence regardless of the nm distance detection that PLA provides. One confusing issue is how the authors use the term 'ultrastructure'. The term should be removed because it is misleading, unless they can provide actual EM evidence. In the absence of stronger imaging data, stating that a-synuclein "accumulates within the lumen of neuronal lysosomes" in the model is not supported by the data. A typical bilayer is ~ 4-5nm in width, and PLA cannot determine if aggregates are inside the lysosome, on surface, or adjacent in the cytosol. The quality and resolution of the PLA data can also be improved.

We have now added the requested immuno-electron microscopy to directly visualize α Syn aggregates in the lysosomal lumen (New Fig. 3b).

We have also quantified the proximity ligation experiment (New Fig. 3e), and removed "ultrastructurally" from the description due to the caveat of antibody size vs. the bilayer thickness.

In addition to the experiments suggested by this reviewer, we have also performed limited proteolysis on immunisolated lysosomes in the presence and absence of detergent, confirming that α Syn aggregates are indeed located in the lysosomal lumen, and are not peripherally associated (New Fig. 3c).

These changes are in conjunction with previous biochemical data using various orthogonal techniques:

- a) Enrichment of dextranosomes containing α Syn aggregates and the lysosome luminal enzyme Cathepsin-L. We confirmed intact lysosomes via a Cathepsin-D activity assay, and protein markers show that our enrichment was clean (Fig. 1).
- b) Immunocapture of lysosomes containing α Syn aggregates and the lysosome luminal enzyme Cathepsin-L from mouse brains and primary neuron cultures using HA LAMP1^{Myc} and TMEM192^{3xHA}. Protein markers show that our isolation worked (Fig. 3a, S6).
- c) Using $^{APEX-2}$ LAMP1 to specifically label neuronal lysosome luminal contents containing α Syn aggregates and the lysosome luminal enzyme Cathepsin-L (Fig. 3d).

Regarding the dot-blots, the preponderance of dot blots in the literature does not include membrane-matched loading controls (some examples include Fontaine et al. EMBO 2016, Colla et al. J Neurosci 2012, Guo et al. Brain 2020, and so on), or even an immunoblotted loading control, which we however show. We would like to note that dot blots, if not performed for a protein of a molecular mass that cannot be fully resolved by SDS-PAGE (e.g. SDS-resistant aggregates), are only of limited value due to the lack of information whether the quantified species is of the right molecular mass. We had therefore chosen to include the more specific, size-separated loading controls, using SDS-PAGE followed by immunoblotting.

Importantly, in our study, dot blots are never alone in any of our figures (Figs. 1a, 3a, 3c, 3d, 4a, 4b, 5a, 5b, 6c, 6d, S1, S4f, S5a, S5c, S5d, S6, S8a, S8b, S10b). The dot blots are always accompanied by immunoblots of aggregates probed with either α Syn, pSer129 α Syn, or myc antibodies (for myc- α Syn *in vitro* experiments). The α Syn aggregates detected by immunoblotting all correlate very well with the quantifications of the filamentous and amyloid-type α Syn detected in the dot blots. In addition, similar to the immunoblots, for all dot blots, we included multiple independent experiments.

Yet, to satisfy this reviewer's concern, we have now also added membrane-matched loading controls to the dot blots performed on lysates or isolated organelles/fractions in the main figures (New Figs. 1a, 3a, 3d, 5a-b, and 6d).

2) There was a request to examine endogenous wt α -synuclein. Perhaps the reasoning for this request was not clear. While it is obvious that WT mice do not have α -synuclein aggregates, the evidence that this pathway is specific for exocytosis of aggregated α -synuclein and is involved in disease pathogenesis is lacking. It is well established that α -synuclein monomers are processed through lysosomes, and that exocytosis of physiological forms occurs in primary cultures (eg Lee et al J Neurosci 2005).

We respectfully disagree with the reviewer. The paper cited (Lee et al J Neurosci 2005) surprisingly has no quantifications because of $n = 1$, and therefore, there is no evidence of reproducibility. More importantly, this study proposes α Syn in large dense core vesicles (LDCVs) or possibly in crude synaptic vesicle fractions which may contain LDCVs, but not in lysosomes. There is no mention of lysosomes anywhere in the paper, and LDCVs are very different from lysosomes (Courel et al. 2010; Winkler et al. 1998; Adalberto Merighi 2018). Therefore, this study is not relevant to our findings here, and we had included it in the introduction only to refer to exocytosis studies of other types of vesicles.

We would also like to note that the localization of α Syn in LDCV lumen is certainly not well established. Physiologically, α Syn is most commonly found as a cytosolic pool that is in equilibrium with an alpha-helically folded form bound to the cytoplasmic leaflet of synaptic vesicles. Other studies have identified α Syn in the nucleus (Maroteaux et al. 1988), in mitochondria (Devi et al. 2008), and in other locations with less clear evidence of a physiological or pathological role. Our study focuses on α Syn aggregates in lysosomes, and we agree with this reviewer that α Syn monomers are shown to be processed through lysosomes, and that subsequently, these monomers are likely to be rapidly turned over, in particular because of the natively unstructured nature of α Syn. We had highlighted in our previous reply that a natively unfolded peptide/protein is not likely to have a significant steady-state presence in the lysosomal lumen due to a short half-life (e.g. Suppl. Figs S3b and S4a in Cuervo et al., 2004). We have shown in both brains and cultured neurons that the endogenous WT and the overexpressed A53T α Syn monomers are neither present within lysosomes (Figs. 1, 2a, 2c, 2d, S6) nor in the media (Figs. 5a-b, S5d, S8a-b).

More importantly, there is a concern that the artificial overexpression by the TG mice, which is very unnatural, causes irrelevant phenotypes. The release of α -synuclein could occur simply because there is too much protein expressed.

Overexpression is a very conventional tool to achieve neuronally expressed α Syn to form aggregates, using either transgenic animals (Kahle et al., J. Neurosci 2000; van der Putten et al., J. Neurosci 2000; Masliah et al., Science 2000; and reviewed in Fernagut & Chesselet, Neurobiol Dis 2004), viral overexpression (reviewed in Huntington TE et al. Aging Dis. 2021), with directional support from duplications and triplication of the *SNCA* gene directly causing PD (Singleton et al. Science 2003; Chartier-Harlin et al. Lancet 2004; Farrer et al. Ann Neurol 2004; Ibanez et al. Lancet 2004; Ahn et al., Neurology 2008), or even exogenously added purified fibrils as a special case of super-physiological expression required to produce aggregates.

For example, the study cited by the reviewer (Lee et al. 2005) used differentiated SH-SY5Y cells followed by a seemingly massive adenoviral overexpression of α Syn: Please see their Figs 1A-B (also below) for a comparison of the overexposure needed to detect the miniscule amount of monomer in the media.

We would also like to note that we have avoided early testing of cultured neurons (unlike the DIV7 and 9 analysis in Lee et al, J Neurosci 2005) because until neurons mature and the glial layer is consolidated, cell death is common and may possibly lead to the detection of α Syn monomers in the medium. Only after neurons and their synaptic connections have matured at >DIV12, the neuronal population becomes stable. Also, α Syn does not localize to synapses and post-synaptic morphology does not mature (i.e., mostly “headless” spines) until weeks 2-3. Per the reviewers’ suggestions, we had included controls for cell death in the previous revision, which has strengthened our study (Fig. S5e-f, S9d).

Importantly, in our Tg^{x2}-αSyn^{A53T} primary neuron model, the same amount of overexpression is present at the earlier time points and in the later time points, as seen by the αSyn monomer levels present in the total cell lysate. However, there are no aggregates present at the early time points, and thus no aggregates are released into the media (Fig. S5a total cell lysate DIV35 versus DIV49; Fig. S5d culture media DIV35 versus DIV49). Therefore, the release of αSyn is not simply occurring because there is too much protein expressed.

Showing an n=1 lyso-IP of 1 month A53T in Fig 2A, and DIV 21 in Fig S6 with no quantifications or evidence of reproducibility, are very weak controls and do not alleviate this concern.

We beg to differ here. Fig. 2a (now Fig. 3a) includes n=8, which was previously stated in the figure legend. We had also previously quantified the fraction of αSyn monomers and aggregates in neuronal lysosomes for this experiment (see bar graph). Fig. S6 includes n=5. Please note that there are no αSyn monomers or aggregates in αSyn^{-/-} culture and no αSyn aggregates are present in Tg^{x2}-αSyn^{A53T} cultures at the earlier time points. Quantification of αSyn aggregates in these conditions would not be accurate because ~0 of ~0 gives absurd, very high or negative, numbers. Therefore, quantifications for these particular experimental conditions were forgone. Yet, quantification in the Tg^{x2}-αSyn^{A53T} neurons at DIV49 is possible, and we have now added the quantification of monomers and aggregates found in isolated lysosomes from the Tg^{x2}-αSyn^{A53T} neurons at day 49 (Fig. S6).

There are several additional missing specificity controls that are required. For example, does dramatic overexpression of an unrelated lysosomal substrate also lead to its exocytosis? Do A53T mice exhibit increased lysosomal exocytosis of other substrates, compared to non-Tg mice?

Cathepsin-L, a native lysosomal protein which is not overexpressed, is also lysosomally exocytosed in WT neurons (Suppl. Fig. S9b&c) and reduced by SNAP23^{KD} and VAMP7^{DN}. Therefore, no overexpression is necessary for this basic cellular process, and this had been previously proposed in other cell types (Andrews NW. Trends Cell Biol. 2000).

Most importantly, in our Tg^{x2}-αSyn^{A53T} primary neuron model, the same amount of overexpression is present at the earlier time points and in the later time points as seen by the αSyn monomer levels present in the total cell lysate. However, there are no aggregates present at the early time points, and thus no aggregates are released into the media (Fig. S5a total cell lysate DIV35 versus DIV49; Fig. S5d culture media DIV35 versus DIV49). Therefore, the release of αSyn is not simply occurring because there is too much protein expressed.

As an additional control, we have now added Tg^{x2}-αSyn^{A53T} neuron data (Suppl. Fig. S9 b&c), showing that overexpression of αSyn does not change lysosomal exocytosis of Cath-L and its sensitivity to SNAP23^{KD} and VAMP7^{DN}. This was an informative control, and we thank the reviewer to pointing it out.

Last, we did not propose that lysosomal exocytosis is specific to αSyn aggregates. Other lysosomal contents such as Cathepsin-L are also released, and to the same levels from WT and Tg^{x2}-αSyn^{A53T} neurons (Suppl. Fig. S9 b&c). It is in fact a very interesting question to pursue if αSyn aggregates modulate lysosomal exocytosis.

Does endogenous a-synuclein (aggregated or not) travel through the same pathway?

It seems that the authors have a good system to address the fate of endogenous wt a-synuclein that accumulates in the CSP^α^{-/-} model, but this was not attempted. Without these data, the relevance and novelty of these studies is limited.

We did not detect monomeric mouse or human αSyn in lysosomes (Fig. 1a; Fig. 3a; Fig. 3c-d; Fig. S6), likely due to the reasons stated above.

Endogenous monomeric αSyn does not accumulate in CSP^α^{-/-} brains (Chandra et al. Cell 2005; and Suppl. Fig. S1). The CSP^α^{-/-} mouse dies in <60 days and the mouse is only rescued by transgenic αSyn overexpression (Chandra et al. Cell 2005). Thus, this mouse is not suited for studying endogenous mouse αSyn. We are happy to share the CSP^α^{-/-} or the rescued CSP^α^{-/-}/Tg-αSyn^{A53T} mouse lines to anyone who is interested in studying the fate of endogenous mouse WT αSyn. However, studying the fate of endogenous

mouse WT α Syn is not the topic of this study, since we focus here on the release mechanism of α Syn aggregates generated by neurons.

3) In vitro aggregation assays in Figure 5 remain poorly controlled. There is no molecular crowding control, and no a-synuclein- immunodepleted control. The use of SNCA $-/-$ media doesn't alleviate this concern, since other non- a-synuclein substrates also likely accumulate and are released from lysosomes in this model, and could promote aggregation through cross seeding, molecular crowding or other mechanisms. If seeds are truly present, agitation is not needed (Buell et al PNAS 2014), and other chemical additives are would certainly confound the results further.

We thank the reviewer for the suggestion and agree that chemical additives would not be a good addition; hence, we did not use any. We would like to respectfully emphasize that the findings of Buell et al., 2014 cannot be directly compared to our study. In the 2014 study, Buell & colleagues use highly concentrated, recombinant purified α Syn seeds modified by sonication, while we are using media containing exocytosed α Syn aggregates from primary neurons that were generated within the neurons. The findings of Buell et al. suggest that highly concentrated synuclein PFFs, generated in bacteria and modified by sonication to amplify seeding surfaces, may template aggregation without agitation; however, in their study, lower concentrations of seeds did not significantly increase aggregation unless exposed to 1) agitation, 2) non-physiological temperature (40°C) or 3) or acidic pH (below 6).

We used myc-tagged recombinant α Syn (myc- α Syn) as a readout for in vitro aggregation kinetics to distinguish it from the untagged α Syn seeds produced by neurons and secreted into the media. All the in vitro conditions were the same except for the media added (Fig. 6 a-c and Fig. S10b). We found that the recruitment of monomeric myc- α Syn into aggregates was much faster when media collected from the Tg^{x2}- α Syn^{A53T} culture (between DIV47-49) was added. α Syn^{-/-} and WT controls were added at the request of reviewers in the last round of revisions. Both, α Syn^{-/-} and WT neurons, do not release α Syn monomers and aggregates into the culture media (Fig. S5d). Tg^{x2}- α Syn^{A53T} cultures at DIV35 express the same amount of α Syn monomers as DIV49 (Fig. S5a), but do not yet accumulate α Syn aggregates in the lysosome and thus do not release these α Syn aggregates into the media (Fig. S5d). Altogether, this indicates that the key player that is released into the media are the α Syn aggregates from Tg^{x2}- α Syn^{A53T} DIV49 neuron cultures.

While α Syn levels are different between α Syn^{-/-}, WT, Tg- α Syn^{A53T}, and Tg^{x2}- α Syn^{A53T} neuron cultures, the total protein level in their medium is similar (*quantifications shown below*), simply due to the large amount of non- α Syn proteins and peptides secreted by neurons and glia – from more than 100 known neuropeptides to larger secreted proteins like BDNF, GDNF, neuroserpin, etc., to cleaved extracellular fragments of transmembrane proteins such as APP, neuexins, etc. All in all, a large number of proteins identifiable by secretomic studies are secreted by neurons and glia combined. While crowding differences and major effects of another unknown non- α Syn substrate can be considered “possible”, they are unlikely due to the overwhelming amount of α Syn genotype-unrelated proteins in the media. It is interesting if there are small changes to the media components between the genotypes, but quantifying any or every differences in the secretome of the cultures is beyond the scope of this study.

primary neuron culture (DIV49)

Primary neuron culture (DIV14)

Response to reviewer figure. Total protein concentration in neuron culture media quantified by BCA assay. (left panel) n=3 pups per genotype. None of the genotypes are significantly different by 1-way ANOVA and by non-parametric Kruskal-Wallis test. (right panels) n=4 cultures per condition. These total protein concentrations were used to normalize Cath-L and neuroserpin levels in Suppl. Fig. S9b-c. None of the protein levels are significantly different from WT control by 1-way ANOVA with Dunnett's multiple-comparison correction or by non-parametric Kruskal-Wallis test with Dunn's multiple-comparison adjustment.

Further, we had also included a 'no medium' PBS control and an 'unconditioned media' control (Suppl. Fig. S10b), meant to eliminate any background or crowding effect between α Syn^{-/-} media, WT media, and Tg^{x2}- α Syn^{A53T} media. There was no difference in aggregation kinetics of recombinant myc- α Syn when no media, unconditioned media, α Syn^{-/-} media, or WT media were added (Suppl. Fig. S10b).

Lastly, but very importantly, we have also shown propagation of aggregation in primary neurons, which is independent and complementary to the cell-free study, and is not sensitive to the same background effects as the aggregation kinetics of purified α Syn (Fig. 6d).

4) In figure 5C, the authors inappropriately equate the HMW bands with filamentous amyloid-type aggregates. There is no evidence to support these structural statements from a western blot analysis.

We agree and apologize for any misunderstanding. If the reviewer is referencing "When myc- α Syn aggregation kinetics were measured by quantitative immunoblotting for either disappearance of myc- α Syn monomers **or for the appearance of filamentous and amyloid-type aggregates**, we again found that compared to the wild type neuron medium, aggregation was augmented by the Tg^{x2}- α Syn^{A53T} medium, and the expression of VAMP7^{DN} fragment diminished this effect" in the results section, we did not measure the HMW bands of the immunoblot of the recombinant myc- α Syn. We instead measured the disappearance of the myc-tagged α Syn monomers since the monomers are recruited into aggregates, and separately measured the appearance of filamentous and amyloid-type aggregates by the dot blots below the myc-tagged α Syn immunoblot. We have clarified this better now.

Please note that the antibodies used have been previously characterized and are well-used by the field:

The α Syn filamentous (α Syn^{Fila}) antibody is a recombinant antibody: Clone MJFR-14-6-4-2 (sold by Abcam, ab209538) produced and characterized in collaboration with the Michael J. Fox foundation.

Figure below is from the vendor's website (Abcam), provided by Dr. Poul Henning Jensen (Aarhus University and The Danish Research Institute of Translational Neuroscience), produced in collaboration with Michael J. Fox Foundation:

Dot Blot showing the reactivity of ab209538 (2 ng/ml) with

F: alpha synuclein Filament

F+FA: alpha synuclein Filament treated in 50% formic acid for 1 h 37oC prior to application to the dot blot.

M+FA: alpha synuclein Monomer treated in 50% formic acid for 1 h 37oC prior to application to the dot blot.

Loading control antibody (1:1000) reacts with Alpha-synuclein irrespectively of it being in a filament, oligomer or a monomer.

This antibody has been used in at least 33 publications other than our manuscript, seen on the vendor's website: <https://www.abcam.com/alpha-synuclein-aggregate-antibody-mjfr-14-6-4-2-conformation-specific-ab209538.html>

A11 antibody, raised to "synthetic molecular mimic of soluble oligomers", recognizes amino acid sequence-independent amyloid oligomers of proteins or peptides folded in the beta-sheet rich amyloid folds. A11 appears

to recognize a peptide backbone epitope common to amyloid oligomers, but not found in native proteins, amyloidogenic monomer or mature amyloid fibrils. A11 has been used to recognize amyloid-type oligomeric species of many polypeptides including A β 40 and 42, human insulin, prion protein, polyglutamine, lysozyme, tau, alpha-synuclein, yeast prion Sup35, alpha B crystallin etc. Some publications showing these (18 with western blots) are available on the vendor's website:

<https://www.thermofisher.com/antibody/product/AHB0052.html>. *Note:* Although A11 is proposed to recognize amyloid-like oligomers and not mature fibrils, we do not use the term "oligomer" throughout the paper, since there is no reliable/clear biophysical reason given that a conformation-specific antibody will react only with an oligomer but not with a larger aggregate of the same/similar conformation.

Finally, we have not used either of these antibodies alone in any figure to derive a conclusion:

Signals from both of these antibodies correlate tightly with SDS-resistant aggregates observed with α Syn and α Syn^{pSer129} antibodies (Figs. 1a; 3a; 3c; 3d; 4a-b; 5a-b; 6d and Suppl. Figs. S1; S4f; S5a; S5c; S5d; S6; S8 a-c). α Syn^{pSer129} is well-characterized as a marker for pathological α Syn species. These four complementary antibodies, each with distinct advantages and caveats, have been put together to arrive at conclusions.

Even further, signals from either of these antibodies also correlate tightly with the fluorescence signal of conformation-specific dyes Thioflavin-T and the Congo Red analog K114 (Fig. 6a-c).

5) There is a novelty issue regarding the conceptual advance. Much work on a-synuclein in lysosomes / transfer has been explored, and recent publications already have described a mechanistic role of SNAP23 and VAMP7 in the exocytosis of a-synuclein (Zhao et al, Mol Neurobio 2021). Lysosomal accumulation of pathogenic a-synuclein has been shown in multiple studies including Lee HJ et al J Neurosci 2005, and Senol AD et al PLOS Bio 2021 both showing direct ultrastructural evidence for a-synuclein inside lysosome. Lysosomal exocytosis has been previously shown as a release mechanism, some with overexpression of a-synuclein, and others with endogenous and more relevant WT expression (Zhao et al 2021; Senol AD, et al 2021; Tsunemi et al J. Neurosci 2019).

We appreciate the reviewer's concern regarding the novelty of our study. We have carefully reviewed each reference. Below is a summary of key differences between the referenced studies and our work, underscoring the novelty factor of our study.

As mentioned in our reply to point 2 above, Lee HJ et al., J Neurosci 2005 do not show, or even claim to show any evidence of α Syn localization in lysosomes. Instead, Lee et al. show α Syn localization to LDCVs (large dense core vesicles) and synaptic-like vesicles. Please see also our response to point 2 for further discussion of this paper. The referenced study by Zhao et al., 2021 contradicts these findings and claims that α Syn is not found in LDCVs in cultured primary neurons. This discrepancy further underlines the importance of our study to delineate a clear mechanistic pathway for α Syn aggregate release from neurons.

Senol AD et al., PLOS Biol 2021 feed exogenous Alexa-labeled α Syn fibrils to undifferentiated CAD cells, which, like the classical experiments of feeding latex beads to cells, will inevitably be trafficked from the plasma membrane to lysosomes (e.g. Chow et al. 2004 *Current Protocols in Cell Biol*). Unlike our study, Senol AD et al's recombinant α Syn fibrils are not neuronally produced α Syn aggregates. Senol et al. feed exogenous fibrils to the donor cell and then show their movement through the cell to cytosolically-connected acceptor cells via tunneling nanotubes. Their proposed pathway does not involve the release of neuronally generated α Syn aggregates into the extracellular space.

Tsunemi et al., 2019 show a buildup of intracellular monomeric α Syn (α Syn shown is ~14kDa). They propose that promoting release of this monomeric α Syn is beneficial for neuron health, whereas our study shows lysosomal exocytosis of SDS-resistant α Syn aggregates as the release-mechanism for seeding-competent aggregates. Importantly, Tsunemi et al. do not show accumulation of α Syn aggregates specifically within lysosomes. Instead, they show that PARK9 mutations that reduce lysosomal exocytosis (measured by activity of lysosomal hydrolases in the culture media) may also lead to increased accumulation of monomeric α Syn within that cell. Again, there is no investigation of aggregated α Syn in this study.

Zhao et al., 2021 did not explore a mechanism for release of pathogenic α Syn aggregates. Instead, Zhao et al. find that monomeric α Syn secretion is "mediated by multiple vesicular pathways including exocytosis of recycling endosomes, multivesicular bodies, autophagosomes, and lysosomes" via an shRNA knockdown study (with no specific rescue of knockdowns, thereby lacking controls for specificity and off-target effects),

which does not clarify the current standing of the field as stated in our introduction. Furthermore, STED imaging, as used by Zhao et al., does not differentiate between luminal/cytosolic membrane association with *any* vesicle, a concern regarding our work which was previously mentioned by Reviewer 2. In contrast, we provide evidence via multiple, complementary techniques that α Syn aggregates are localized to the lysosomal lumen: electron microscopy, APEX-2 proximity labeling, proximity ligation assay and limited proteolysis of immunisolated neuronal lysosomes. Besides, Zhao et al., 2021 was first received on May 26, 2021 and published on October 12, 2021. Our manuscript was first received on March 3, 2021, and reviewers gave us feedback that we received from the editor on March 29, 2021, inviting us for revisions. This was all before Zhao et al.'s publication. Following the initial positive addressable feedback from the reviewers, our manuscript was uploaded as a preprint on BioRxiv on April 11, 2021 which included the co-IP screen to verify the lysosomal SNAREs required for exocytosis in mouse brain and the two strategies to target the SNAREs involved to disrupt exocytosis of α Syn aggregates from neurons: shRNA knockdown of mouse SNAP23 with rescue via expression of human SNAP23 cDNA and expression of the VAMP7^{DN} fragment: (<https://www.biorxiv.org/content/10.1101/2021.04.10.439302v1.article-info>).

Overall, the field has not yet reached a consensus on the mechanism of release for pathogenic α Syn aggregates or other prominent aggregates in neurodegeneration from neurons, hence the ambiguity in this recent review and the vague cartoon depicted in Fig. 1 (Peng et al. Nature Reviews Neurology 2020). Although there remains much more work to be done, our study is novel because we have generated new mouse and primary neuron α Syn aggregation models that have uniquely equipped us to parse apart the cellular-molecular process involved in the release of α Syn aggregate from neurons. Using our models, we are the first to directly isolate neuronal lysosomes containing endogenously produced α Syn aggregates and are the first to trace the release of α Syn aggregates from neuronal lysosomes into the extracellular space. Importantly, none of the papers cited above have shown data from neuronally generated α Syn aggregates.

The intro should include a better description of the link between a-synuclein and lysosomes. The authors indicate that the csp^{-/-} a-synuclein data in the first part of the results 'provided a clue' that a-synuclein could accumulate in lysosomes, when in fact there is already a large amount of published data suggesting this. For example, synuclein is a common pathology seen in many lysosomal disorders (Shachar et al Mov. Dis. 2011). Multiple experimental studies have shown links between synuclein and lysosomes.

The study-initiating CSP α ^{-/-} data are from 2014, and they did provide us with the clue for our study. As a small early-stage lab, it has taken a while to finish this work which involved transgenic mouse generation, crossing, aging, and developing a new primary culture model suitable to address the question of release of neuronally produced α Syn aggregates. However, to avoid any confusion and respond to this reviewer's concern, we have now removed "provided a clue", and changed the sentence to include the studies showing association between lysosomal dysfunction/storage with PD (including Shachar et al. 2011).

We have now also included in the introduction the link between Parkinson's disease and lysosomal disorders citing Shachar et al. 2011, and other studies.

Reviewer #3 (Remarks to the Author):

The authors have adequately addressed all my comments, and this revised version is significantly improved. I have no further concerns.

Sincere thanks to the reviewer for taking valuable time to review our work.

REVIEWERS' COMMENTS

Reviewer #1 (Remarks to the Author):

The authors have addressed all previous concerns.

In the present version of the study, the addition of the immuno-EM evidence of aggregated/filamentous alpha-synuclein inside lysosomes (Fig. 3b) is important, and further strengthens the conclusions of the study. However, the presentation can be improved, and some additional details are needed. Specifically, in Figure 3b, the authors should include micrographs from larger cytoplasmic areas, omit the added white/red border on the lysosomal limiting membrane. In addition, according to the quantification, a representative image of a WT lysosome with gold particles should be included, and the finding that also some WT lysosomes appeared positive for aSyn aggregated/filamentous forms in EM can be briefly discussed. Within the Results section text, it should be clarified that these are neuronal lysosomes. In the corresponding Methods section, the approximate anatomical coordinates of the cerebral cortex in sections examined by immune-EM, and the criteria used for identifying lysosomes within a neuron, as opposed to a glial cell, must be added.

Reviewer #2 (Remarks to the Author):

the authors have done a nice job in the reply and all concerns have been addressed.

REVIEWERS' COMMENTS

Reviewer #1 (Remarks to the Author):

The authors have addressed all previous concerns.

We would like to thank the reviewer for her/his valuable time, and insightful suggestions throughout the review process.

In the present version of the study, the addition of the immuno-EM evidence of aggregated/filamentous alpha-synuclein inside lysosomes (Fig. 3b) is important, and further strengthens the conclusions of the study. However, the presentation can be improved, and some additional details are needed. Specifically, in Figure 3b, the authors should include micrographs from larger cytoplasmic areas, omit the added white/red border on the lysosomal limiting membrane. In addition, according to the quantification, a representative image of a WT lysosome with gold particles should be included, and the finding that also some WT lysosomes appeared positive for aSyn aggregated/filamentous forms in EM can be briefly discussed. Within the Results section text, it should be clarified that these are neuronal lysosomes. In the corresponding Methods section, the approximate anatomical coordinates of the cerebral cortex in sections examined by immune-EM, and the criteria used for identifying lysosomes within a neuron, as opposed to a glial cell, must be added.

Fig. 3b has now been edited as suggested, and all the reviewer's other suggestions have now been included in the text of the manuscript. These were constructive points and improved the EM figure, and we thank the reviewer for the suggestions.